# COMMUNICATION-EFFICIENT FEDERATED LEARNING WITH ACCELERATED CLIENT GRADIENT

## ABSTRACT

Federated learning often suffers from slow and unstable convergence due to the heterogeneous characteristics of participating client datasets. Such a tendency is aggravated when the client participation ratio is low since the information collected from the clients has large variations. To address this challenge, we propose a simple but effective federated learning framework, which improves the consistency across clients and facilitates the convergence of the server model. This is achieved by making the server broadcast a global model with a gradient acceleration. This strategy enables the proposed approach to convey the projective global update information to participants effectively without additional client memory for storing previous models, and extra communication costs. We also regularize local updates by aligning each client with the overshot global model to reduce bias and improve the stability of our algorithm. We provide the theoretical convergence rate of our algorithm and demonstrate remarkable performance gains in terms of accuracy and communication efficiency compared to the state-of-the-art methods, especially with low client participation rates. We plan to release our code to facilitate the reproduction of our work.

## 1 INTRODUCTION

Federated learning (FL) (McMahan et al., 2017) is a large-scale machine learning framework that learns a shared model in a central server through collaboration with a large number of remote clients with separate datasets. Such a decentralized learning framework achieves a basic level of data privacy because the data stored in local clients are unobservable by the server and other clients. On the other hand, federated learning algorithms are particularly sensitive to communication and computational costs because many clients such as mobile or IoT devices generally have limited resources.

A baseline algorithm of federated learning, FedAvg (McMahan et al., 2017), updates a subset of its client models based on a gradient descent method using their local data and then uploads the resulting models to the server for estimating the global model parameters via model averaging. As discussed extensively on the convergence of FedAvg (Stich, 2019; Yu et al., 2019; Wang & Joshi, 2021; Stich & Karimireddy, 2019; Basu et al., 2020), multiple local updates conducted before server-side aggregation provide theoretical support and practical benefit of federated learning by reducing communication costs significantly.

Despite the initial success, federated learning faces two key challenges: high heterogeneity in training data distributed over clients and limited participation rates of clients. Several studies (Zhao et al., 2018; Karimireddy et al., 2020) have shown that multiple local updates in the clients with non-*i.i.d.* (independent and identically distributed) data lead to client model drift, in other words, diverging updates in the individual clients. Such a phenomenon introduces the high variance issue in FedAvg steps for global model updates, which hampers the convergence to the optimal average loss over all clients (Li et al., 2020a; Wang et al., 2019b; Khaled et al., 2019; Li et al., 2020b; Hsieh et al., 2020; Wang et al., 2020). The challenge related to client model drift is exacerbated when the client participation rate per communication round is low, due to unstable client device operations and limited communication channels.

To properly address the client heterogeneity issue, we propose a novel federated learning algorithm, Federated averaging with Accelerated Client Gradient (FedACG), which conveys the momentum of the global gradient to clients and enables the momentum to be incorporated into the local updates in

the individual clients. Specifically, FedACG transmits the global model integrated with the global momentum in the form of a single message, which allows each client performs its local gradient step along the landscape of the global loss function. This approach turns out to be effective in reducing the gap between global and local losses. In addition, FedACG adds a regularization term in the objective function of clients to make the local gradients more consistent across clients. We show that subtle differences of federated learning algorithms can make a significant impact on final results and discuss the behavior of FedACG together with related methods.

The main contributions of this paper are summarized as follows.

- We propose a simple but effective federated optimization algorithm, called FedACG, that employs the global momentum to initialize client models for accelerated updates and aligns the gradients of individual clients with that of the server .
- FedACG is free from additional communication cost, extra computation of the server, and memory overhead of clients; these properties are desirable for the real-world settings of federated learning.
- We show that the FedACG algorithm is theoretically sound by providing its formal convergence guarantee and rate.
- FedACG demonstrates outstanding performance in terms of communication efficiency and robustness to client heterogeneity, especially with low participation rate of clients.

## 2    RELATED WORK

Federated learning was first introduced in McMahan et al. (2017). It formulates the problem and provides the FedAvg algorithm as a solution for its key challenges such as non-*i.i.d.* client data, massively distributed clients, and partial participation of clients. Many works explore the negative influence of heterogeneity in federated learning empirically (Zhao et al., 2018) and derive convergence rates depending on the level of heterogeneity (Li et al., 2020a; Wang et al., 2019b; Khaled et al., 2019; Li et al., 2020b; Hsieh et al., 2020; Wang et al., 2020).

There exists a long line of research for client-side optimization to alleviate the divergence of clients from the global model. FedProx (Li et al., 2020a) penalizes the difference between the server and client parameters while FedDyn (Acar et al., 2021) and FedPD (Zhang et al., 2020) adopt cumulative gradients of each client for dynamic regularization of local updates. FedDC (Gao et al., 2022) introduces the auxiliary drift variables for each client to reduce the impact of the local drift on the global objective. There is another line of work that employs variance reduction techniques in client updates to eliminate inconsistent updates across local models. SCAFFOLD (Karimireddy et al., 2020) and Mime (Karimireddy et al., 2021) employ control variates for local updates while FedDANE (Li et al., 2019) adds a gradient correction term based on the server gradient. FedPA (Al-Shedivat et al., 2021) debiases client updates by estimating the global posterior on the client side. On the other hand, some approaches adopt a contrastive loss (Li et al., 2021), knowledge distillation (Kim et al., 2022), or a generative model (Zhu et al., 2021) to ensure the similarity of the representations between the global and local models. FedSAM (Qu et al., 2022) and ASAM (Caldarola et al., 2022) apply SAM (Foret et al., 2021) as a client-side optimizer for reducing the gap between global and local losses. However, most of these methods require full participation (Zhang et al., 2020; Khanduri et al., 2021), additional communication cost (Xu et al., 2021; Karimireddy et al., 2020; Zhu et al., 2021; Karimireddy et al., 2021; Li et al., 2019; Das et al., 2020; Gao et al., 2022), or extra client storage (Acar et al., 2021; Karimireddy et al., 2020; Li et al., 2021; Gao et al., 2022), which are problematic in realistic federated learning tasks.

Momentum-based optimization techniques have also been explored for the stability and speedup of convergence. These approaches incorporate a momentum SGD (Hsu et al., 2019; Wang et al., 2019a) or an adaptive gradient-descent method (Reddi et al., 2021; Caldarola et al., 2022) into the server model update while FedCM (Xu et al., 2021) and FedADC (Ozfatura et al., 2021) employs global momentum for a gradient correction in the local updates. STEM (Khanduri et al., 2021) and Fed-GLOMO (Das et al., 2020) apply the STORM algorithm (Cutkosky & Orabona, 2019) to both server- and client-level SGD procedures for reducing high variance in server model updates. Unlike these methods that require additional communication overhead for transmitting the global momentum for

local updates, FedACG saves the communication cost by broadcasting the momentum-integrated model as a single message.

Meanwhile, another set of works aims to decrease the communication cost per round by compressing the model transmitted. FedPAQ (Reisizadeh et al., 2020), FedCAMS (Wang et al., 2022), and FedCOMGATE (Haddadpour et al., 2021) quantize the communicated message by using low-bit precisions while FedPara (Nam et al., 2022) adopts a low-rank Hadamard product to reparameterize the model's weights. These works are orthogonal to ours and can be integrated into our algorithm.

## 3 PROPOSED APPROACH

### 3.1 PRELIMINARIES

**Problem setting and notations** Let $\mathcal{L}_i(\theta) := \mathbb{E}_{(x,y) \sim \mathcal{D}_i}[\ell_i((x, y); \theta)]$ be the loss function of the $i^{\text{th}} \in \{1, \ldots, N\}$ client with a local dataset denoted by $\mathcal{D}_i$. Then, our goal is to train a model that minimizes the average loss of all clients as follows:

$$\min_{\theta} \left\{ \mathcal{L}(\theta) := \sum_{i=1}^{N} \omega_i \mathcal{L}_i(\theta) \right\}, \tag{1}$$

where $\theta$ is the parameter of the global model and $\omega_i$ is the normalized weight of the $i^{\text{th}}$ client proportional to the size of the local dataset. We focus on the non-*i.i.d.* data setting, where local datasets have heterogeneous distributions. Note that the communication of raw data between clients and the server is strictly prohibited in principle due to privacy.

**FedAvg algorithm** FedAvg (McMahan et al., 2017) is a standard solution of federated learning, where the server simply aggregates all the participating client models to obtain the global model. Specifically, in the $t^{\text{th}}$ communication round, the server broadcasts the latest global model, represented as $\theta^{t-1}$ to the active clients in $\mathcal{S}_t \subseteq \{1, \ldots, N\}$. Each participating client optimizes its local model by using the global model as its initial point, *i.e.*, $\theta_{i,0}^t := \theta^{t-1}$. After $K$ iterations of the local training, each client uploads its local updates $\Delta_i^t := \theta_{i,K}^t - \theta_{i,0}^t$ to the server, and then the server aggregates them as $\Delta^t := \sum_{i \in S_t} \omega_i \Delta_i^t$. The server constructs the next server model $\theta^t := \theta^{t-1} + \Delta^t$ for the broadcasting in the next round.

Due to non-*i.i.d* data and limited client participation rate in each round of training, FedAvg suffers from client drift (Karimireddy et al., 2020). Such a phenomenon results in the inconsistent updates of client models caused by overfitting to local data of individual clients and consequently leads to the high variance of the global model. This tendency is aggravated over multiple communication rounds in FL because each client initializes its parameters using the global model.

### 3.2 FEDERATED AVERAGING WITH ACCELERATED CLIENT GRADIENT (FEDACG)

To reduce the inconsistency between the local models and the divergence of the resulting global model, we incorporate the global momentum into the local models for guiding local updates.

**Accelerated client gradient** The main idea of FedACG is to revise the initialization of client models using the global model integrated with a global gradient, allowing more effective and stable updates of local models. Since direct computation of the global gradient is impractical in FL, we utilize the global momentum $m^{t-1}$ as a viable approximation, which is updated as $m^{t-1} := \lambda m^{t-2} + \Delta^{t-1}$, at each round. Specifically, in the $t^{\text{th}}$ communication round, the server augments the recent global model $\theta^{t-1}$ with the global momentum $m^{t-1}$. As illustrated in Figure 1, The server then broadcasts this accelerated global model, represented as $\theta^{t-1} + \lambda m^{t-1}$, as a single message to the active clients in $\mathcal{S}_t \subseteq \{1, \ldots, N\}$. Note that $\lambda \in (0, 1)$ controls the importance of the global momentum. Each participating client optimizes its local model from the momentum-integrated initialization. This proactive initialization allows each client to find its local optimal solution along the trajectory of the global gradient, which improves the consistency of local updates in FedACG. Our approach has a similar motivation with meta-learning (Finn et al., 2017), where a meta-learner identifies the optimal point to facilitate the optimization in all target tasks. After $K$ iterations local training, the server

---

**Algorithm 1** FedACG

---

**Input:** $\beta$, $\lambda$, initial server model $\theta^0$, number of clients $N$, number of communication rounds $T$,
       number of local iterations $K$, local learning rate $\eta$
Initialize global momentum $m^0 = 0$
**for** *each round $t = 1, 2, \ldots, T$* **do**
    Sample subset of clients $S_t \subseteq \{1, \ldots, N\}$
    Server sends $\theta^{t-1} + \lambda m^{t-1}$ for all clients $i \in S_t$
    **for** *each client $i \in S_t$,* **in parallel do**
        Initialize local model $\theta^t_{i,0} = \theta^{t-1} + \lambda m^{t-1}$
        **for** *each local iteration $k = 1, 2, \ldots, K$* **do**
            Compute mini-batch loss
            $f_i(\theta^t_{i,k-1}) = \ell_i(\theta^t_{i,k-1}) + \frac{\beta}{2}\|\theta^t_{i,k-1} - (\theta^{t-1} + \lambda m^{t-1})\|^2$
            $\theta^t_{i,k} = \theta^t_{i,k-1} - \eta \nabla f_i(\theta^t_{i,k-1})$
        **end**
        $\Delta^t_i = \theta^t_{i,K} - \theta^t_{i,0}$
        Client sends $\Delta^t_i$ back to the server
    **end**
    **In server:**
        $\Delta^t = \sum_{i \in S_t} \omega_i \Delta^t_i$
        $m^t = \lambda m^{t-1} + \Delta^t$
        $\theta^t = \theta^{t-1} + m^t$
**end**
**Return** $\theta^t$

---

updates its momentum and constructs the next server model $\theta^t := \theta^{t-1} + m^t$ for the preparation of the next round. Algorithm 1 presents the procedure of FedACG.

**Regularization with momentum-integrated model** In addition to the initial acceleration for local training, we augment the client's loss function with the quadratic term $\frac{\beta}{2}\|\theta^t_{i,k} - (\theta^{t-1} + \lambda m^{t-1})\|^2$ which penalizes the difference between the local online model $\theta^t_{i,k}$ and the accelerated global model $\theta^{t-1} + \lambda m^{t-1}$. Note that, $\beta$ controls the intensity of the penalty. The penalized term takes advantage of the global gradient information $\lambda m^t$ to reduce the variations of client-specific gradients, $\Delta^t_i$. This regularization term further enforces the local model not to deviate from the accelerated point, preventing each client from falling into biased local minima.

### 3.3 DISCUSSION

While our formulation has something in common with the existing works that also address client heterogeneity using global gradient information for local updates, FedACG has major advantages. First, contrary to (Karimireddy et al., 2020; Xu et al., 2021; Gao et al., 2022), the server and clients in FedACG only communicate model parameters without imposing additional network overhead for transmitting gradients and other information; the server broadcasts $(\theta^{t-1} + \lambda m^{t-1})$ as a single message and each client sends $\Delta^t_i$ to the server. This is a critical benefit because the increase in communication cost challenges many realistic federated learning applications involving clients with limited network bandwidths. Second, FedACG is robust to the low participation rate of clients and allows new-arriving clients to join the training process without a warmup period because, unlike (Karimireddy et al., 2020; Acar et al., 2021; Li et al., 2021; Gao et al., 2022), the clients are supposed to neither store their local states nor use them for model updates.

**Comparison with FedAvgM** Although FedACG apparently looks similar to FedAvgM in the sense that both methods employ the global momentum for optimization, they have critical differences. To analyze the difference between the two algorithms, we decompose the global model update of FedACG into two steps: 1) updating the previous global model, $\theta^{t-1}$, with the global momentum term, $\lambda m^t$, and compute the interim model $\Phi^t = \theta^{t-1} + \lambda m^{t-1}$, and 2) updating the interim model with the aggregated local gradients, $\Delta^t$, and deriving the new global model, $\theta^t = \Phi^t + \Delta^t$. While FedAvgM performs the local updates from the previous global model, *i.e.*, $\Delta^t := \Delta^t|_{\theta^{t-1}}$, FedACG computes the local updates from the interim model parameter, *i.e.*, $\Delta^t := \Delta^t|_{\Phi^t}$. In other words,

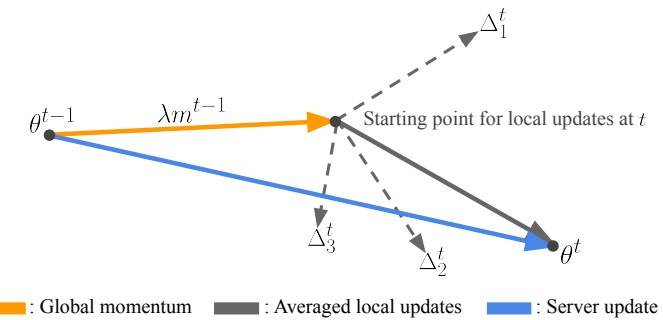

Figure 1: **An illustration of the proposed accelerated client gradient method.** We first partially update the global model in the direction of the global momentum (orange) and then aggregate local updates (gray), resulting in the server model in the next round (blue). This anticipatory update makes individual local updates aligned with the global gradient and achieve speed-up of convergence.

the two algorithms have different initializations, $\theta^{t-1}$ vs. $\Phi^t$, for training local models. As will be discussed in Section 4.3, the incorporation of lookahead initialization allows FedACG to obtain a more robust model than FedAvgM. Such a difference is accumulated over multiple communication rounds and eventually leads to significant performance gaps.

### 3.4 CONVERGENCE ANALYSIS OF FEDACG

We now discuss convergence of the FedACG algorithm in general non-convex FL scenarios. To establish the convergence analysis for FedACG, we make three assumptions; (1) the local loss function $\mathcal{F}_i(\cdot)$ is $L$-smooth, (2) its stochastic gradient $\nabla f_i(x) := \nabla \mathcal{F}_i(x; \mathcal{D}_i)$ is unbiased and possesses a bounded variance, *i.e.*, $\mathbb{E}_{\mathcal{D}_i} \|\nabla f_i(x) - \nabla \mathcal{F}_i(x)\| < \sigma^2$, and (3) the average norm of local gradients is bounded by a function of the global gradient magnitude as $\frac{1}{N} \sum_{i=1}^{N} \|\nabla \mathcal{F}_i(x)\|^2 \leq \sigma_g^2 + B^2 \|\nabla \mathcal{F}(x)\|^2$, where $\sigma_g \geq 0$ and $B \geq 1$. These assumptions are widely used for analyzing the non-convex loss functions in FL in the previous works (Karimireddy et al., 2020; Xu et al., 2021; Acar et al., 2021; Reddi et al., 2021; Karimireddy et al., 2021; Qu et al., 2022). Note that our convergence proof is free from the bounded gradient assumption of the global or local loss while it is commonly used for the proof in momentum-based or adaptive optimization methods (Reddi et al., 2021; Xu et al., 2021; Wu et al., 2023).

We now state the convergence result of FedACG; the detailed proof is provided in Appendix C.

**Theorem 1.** *(Convergence of FedACG for non-convex functions) Suppose that local functions $\{\mathcal{F}_i\}_{i=1}^N$ are non-convex and $L$-smooth. Let $z^t = \theta^t + \frac{\lambda}{1-\lambda} m^t$. Then, by setting $\eta \leq \frac{1-\lambda}{100KL(B^2+1)}$, FedACG satisfies,*

$$\min_{t=1,...,T} \mathbb{E} \left\| \nabla \mathcal{F} \left( \theta^{t-1} + \lambda m^{t-1} \right) \right\|^2 \leq \mathcal{O} \left( \sqrt{\frac{L d_0}{T} \left( \psi_1 \sigma^2 + \psi_2 \sigma_g^2 \right)} \right), \tag{2}$$

*where $d_0 = \mathcal{F}(z^0) - \mathcal{F}(z^*)$, $A = 1 + \frac{4N}{(N-1)|S_t|}(1 - \frac{|S_t|}{N})$, $\psi_1 = \frac{A(2-\lambda)}{K} + \frac{5}{2} + (1-\lambda)(\frac{5}{K} + 2)$, $\psi_2 = A((1-\lambda)^2 + 4) + 10(1-\lambda)$.*

Theorem 1 provides the convergence rate of FedACG, $\mathcal{O}\left(1/\sqrt{|S_t|KT}\right)$, which indicates the linear speedup with respect to $|S_t|$. This rate matches the best convergence rate of existing FL methods (Karimireddy et al., 2020; Acar et al., 2021). Because the coefficient $\psi_2$ decreases as $\lambda$ approaches to 1 from 0, the accelerated client gradient of FedACG reduces the negative impact by the client heterogeneity. Moreover, FedACG is robust to the client's low participation rate because a rise of $\lambda$ reduces the impact of $A$, which is tightly coupled with the client participation rate.

## 4 EXPERIMENTS

This section presents empirical evaluation results of FedACG and competing federated learning methods. Refer to Appendix A for more details about our implementation details with hyperparameter setting, evaluation metrics, and results from ablation studies.

Table 1: Comparisons on CIFAR-10, CIFAR-100, and Tiny-ImageNet with two different federated learning settings. For (a) a moderate-scale experiment, the number of clients and participation rate, are set to 100, and 5%, respectively, while (b) a large-scale setting has 500 clients with a 2% participation rate. The Dirichlet parameter is commonly set to 0.3. Accuracies at the target round and the communication round to reach target test accuracy are based on exponential moving averages with parameter 0.9. The arrows indicate whether higher ($\uparrow$) or lower ($\downarrow$) is better. The best performance in each column is denoted in **bold**. FedCM[†] and FedDC[‡] require 50% and 100% additional communication costs at each communication round, respectively.

(a) Moderate-scale: 5% participation, 100 clients

| Method | CIFAR-10 | | | | CIFAR-100 | | | | Tiny-ImageNet | | | |
| | Acc. (%, $\uparrow$) | | Rounds ($\downarrow$) | | Acc. (%, $\uparrow$) | | Rounds ($\downarrow$) | | Acc. (%, $\uparrow$) | | Rounds ($\downarrow$) | |
| | 500R | 1000R | 81% | 85% | 500R | 1000R | 47% | 55% | 500R | 1000R | 35% | 38% |
|---|---|---|---|---|---|---|---|---|---|---|---|---|
| FedAvg | 74.36 | 82.53 | 840 | 1000+ | 41.88 | 47.83 | 924 | 1000+ | 33.94 | 35.37 | 645 | 1000+ |
| FedProx | 73.70 | 82.68 | 826 | 1000+ | 42.43 | 48.32 | 881 | 1000+ | 34.14 | 35.53 | 613 | 1000+ |
| FedAvgM | 80.56 | 85.48 | 519 | 828 | 46.98 | 53.29 | 515 | 1000+ | 36.32 | 38.51 | 416 | 829 |
| FedADAM | 72.33 | 81.73 | 908 | 1000+ | 44.80 | 52.48 | 691 | 1000+ | 33.22 | 38.91 | 658 | 945 |
| FedDyn | 84.82 | 88.10 | 392 | 646 | 48.38 | 55.79 | 424 | 883 | 37.35 | 41.18 | 344 | 573 |
| MOON | 83.32 | 86.30 | 371 | 686 | 53.15 | 58.37 | 284 | 640 | 36.62 | 40.33 | 410 | 627 |
| FedCM[†] | 78.92 | 83.71 | 624 | 1000+ | 52.44 | 58.06 | 293 | 747 | 31.61 | 37.87 | 694 | 1000+ |
| FedDC[‡] | **86.52** | 87.47 | 323 | 519 | 54.25 | 59.01 | 333 | 553 | 40.32 | 45.51 | 340 | 403 |
| FedACG (ours) | 85.13 | **89.10** | **319** | **450** | **55.79** | **62.51** | **260** | **409** | **42.26** | **46.31** | **226** | **331** |

(b) Large-scale: 2% participation, 500 clients

| Method | CIFAR-10 | | | | CIFAR-100 | | | | Tiny-ImageNet | | | |
| | Acc. (%, $\uparrow$) | | Rounds ($\downarrow$) | | Acc. (%, $\uparrow$) | | Rounds ($\downarrow$) | | Acc. (%, $\uparrow$) | | Rounds ($\downarrow$) | |
| | 500R | 1000R | 73% | 77% | 500R | 1000R | 36% | 40% | 500R | 1000R | 24% | 30% |
|---|---|---|---|---|---|---|---|---|---|---|---|---|
| FedAvg | 58.74 | 71.45 | 1000+ | 1000+ | 30.16 | 38.11 | 842 | 1000+ | 23.63 | 29.48 | 523 | 1000+ |
| FedProx | 57.88 | 70.75 | 1000+ | 1000+ | 29.28 | 36.16 | 966 | 1000+ | 25.45 | 31.71 | 445 | 799 |
| FedAvgM | 65.85 | 77.49 | 753 | 959 | 31.80 | 40.54 | 724 | 955 | 26.75 | 33.26 | 386 | 687 |
| FedADAM | 61.53 | 69.94 | 1000+ | 1000+ | 24.40 | 30.83 | 1000+ | 1000+ | 21.88 | 28.08 | 648 | 1000+ |
| FedDyn | 65.49 | 77.92 | 732 | 936 | 31.58 | 41.01 | 691 | 927 | 24.35 | 29.54 | 483 | 1000+ |
| MOON | 69.15 | 78.06 | 617 | 872 | 33.51 | 42.41 | 601 | 828 | 26.69 | 31.81 | 382 | 741 |
| FedCM[†] | 69.27 | 76.57 | 742 | 1000+ | 27.23 | 38.79 | 872 | 1000+ | 19.41 | 24.09 | 975 | 1000+ |
| FedDC[‡] | 71.86 | **83.49** | 518 | 686 | 34.64 | 45.93 | 569 | 741 | 25.72 | 28.92 | 420 | 1000+ |
| FedACG (ours) | **73.61** | 82.80 | **484** | **605** | **35.68** | **48.40** | **505** | **616** | **31.47** | **38.48** | **246** | **447** |

## 4.1 EXPERIMENTAL SETUP

**Datasets** We conduct a set of experiments on three datasets, CIFAR-10 (Krizhevsky et al., 2009), CIFAR-100 (Krizhevsky et al., 2009), and Tiny-ImageNet (Le & Yang, 2015) with various data heterogeneity levels and participation rates. Note that Tiny-ImageNet (200 classes with $10K$ examples) is more realistic compared to other datasets used for the evaluation of many existing methods (McMahan et al., 2017; Karimireddy et al., 2020). We generate *i.i.d.* data splits by randomly assigning training examples to individual clients without replacement. For non-*i.i.d.* datasets, we simulate the data heterogeneity by sampling the label ratios from a Dirichlet distribution with a symmetric parameter, 0.3 or 0.6, following strategies in Hsu et al. (2019). In both *i.i.d.* and non-*i.i.d.* cases, each client holds the same number of examples as in other works.

We extend our experiments to the widely adopted FL benchmark, LEAF (Caldas et al., 2019), known for its realistic settings. LEAF introduces heterogeneity in class distribution, data quantity, and feature alignment. For FEMNIST, CelebA, and ShakeSpeare, we employ the non-*i.i.d.* data splits provided by this framework.

**Baselines** We compare our method, FedACG, with state-of-the-art federated learning techniques, which include FedAvg (McMahan et al., 2017), FedProx (Li et al., 2020a), FedAvgM (Hsu et al., 2019), FedADAM (Reddi et al., 2021), FedDyn (Acar et al., 2021), FedCM (Xu et al., 2021), MOON (Li et al., 2021), and FedDC (Gao et al., 2022). We adopt the standard ResNet-18 (He et al., 2016) as our backbone network for all experiments after replacing the batch normalization with the group normalization as suggested in Hsieh et al. (2020).

Table 2: Results from a low participation rate (1%) over 500 clients with the Dirichlet (0.3).

| Method | CIFAR-10 | | | | CIFAR-100 | | | |
| | Acc. (%, ↑) | | Rounds (↓) | | Acc. (%, ↑) | | Rounds (↓) | |
| | 500R | 1000R | 64% | 68% | 500R | 1000R | 30% | 35% |
|---|---|---|---|---|---|---|---|---|
| FedAvg | 54.71 | 68.96 | 792 | 949 | 26.94 | 35.69 | 636 | 950 |
| FedProx | 55.18 | 69.80 | 773 | 919 | 26.92 | 35.41 | 648 | 963 |
| FedAvgM | 57.82 | 71.12 | 669 | 812 | 29.29 | 39.36 | 530 | 755 |
| FedADAM | 47.97 | 55.11 | 1000+ | 1000+ | 17.72 | 23.92 | 1000+ | 1000+ |
| FedDyn | 54.86 | 70.78 | 713 | 858 | 27.86 | 36.31 | 595 | 896 |
| MOON | **64.55** | 73.89 | **491** | 645 | 28.29 | 36.37 | 567 | 886 |
| FedCM[†] | 49.21 | 60.38 | 1000+ | 1000+ | 16.32 | 22.59 | 1000+ | 1000+ |
| FedDC[‡] | 60.56 | 75.06 | 610 | 681 | 29.14 | 38.84 | 519 | 789 |
| FedACG (ours) | 63.70 | **76.45** | 497 | **618** | **31.74** | **45.18** | **458** | **581** |

Table 3: Results on CIFAR-100 with dynamic client replacements during training: each client is replaced with a probability of 0.5 at every 100 rounds.

| Method | Acc. (%, ↑) 1000R | Rounds (↓) 38% |
|---|---|---|
| FedAvg | 35.87 | 1000+ |
| Fedprox | 35.89 | 1000+ |
| FedDyn | 38.47 | 941 |
| MOON | 39.57 | 852 |
| FedDC[†] | 36.82 | 1000+ |
| FedACG (ours) | **41.51** | **769** |

Table 4: Contribution of individual components in FedACG. The results are measured after $1K$ round on CIFAR-10 and CIFAR-100 with 2% participation and 500 clients.

| Server update w/ momentum | Accelerated gradient | Local reg. | CIFAR-10 | CIFAR-100 |
|---|---|---|---|---|
| | | | 71.45 | 38.11 |
| | | ✓ | 70.75 | 36.16 |
| ✓ | | | 77.49 | 40.54 |
| ✓ | | ✓ | 76.16 | 44.64 |
| ✓ | ✓ | | 82.20 | 46.80 |
| ✓ | ✓ | ✓ | **82.80** | **48.40** |

## 4.2 MAIN RESULTS

**Evaluation in standard settings**   We first present the performance of the proposed approach, FedACG, on CIFAR-10, CIFAR-100, and Tiny-ImageNet by varying the number of clients, data heterogeneity, and participation rate. Our experiments have been performed in two different settings; one is a moderate scale, which involves 100 devices with a 5% participation rate per round, and the other is with a large number of clients, 500 with a participation rate of 2%. Because the number of clients in the large-scale setting is five times higher than that in the moderate-scale experiment, the number of examples per client is reduced by 80%.

Table 1a demonstrates that FedACG improves accuracy and convergence speed significantly and consistently compared with other federated learning methods in most cases. This is partly because FedACG enables each client to look ahead to the global update and aligns the local model updates with the global gradient trajectory. Note that FedCM and FedDC respectively require $1.5\times$ and $2\times$ communication costs for each communication round since they communicate the current model and the associated gradient information per round while the rest of the algorithms only need to transmit model parameters.

For the large-scale setting, Table 1b illustrates the outstanding performance of FedACG on CIFAR-10, CIFAR-100, and Tiny-ImageNet, except for the accuracy at 1K rounds on CIFAR-10. One noticeable thing is that the overall performance is lower than the case with a moderate number of clients. This is because the number of training data for each client decreases and each client suffers more from the heterogeneous data distributions. Nevertheless, we observe that FedACG outperforms other methods consistently in most cases; the accuracy gap between FedACG and its strongest competitor becomes larger in these more challenging scenarios. The results from the large-scale experiments exhibit the robustness of FedACG to the data heterogeneity and low participation rates of clients. We present more comprehensive results for the convergence of FedACG in Appendix A.5.1.

**Effect of low participation rate**   One of the critical challenges in federated learning is partial participation of clients, which can slow down the convergence of the global model. To verify the robustness of FedACG to low client participation rates, we conduct experiments with 500 clients and a participation rate as low as 1%. Since the numbers of local epochs and iterations are set to 5

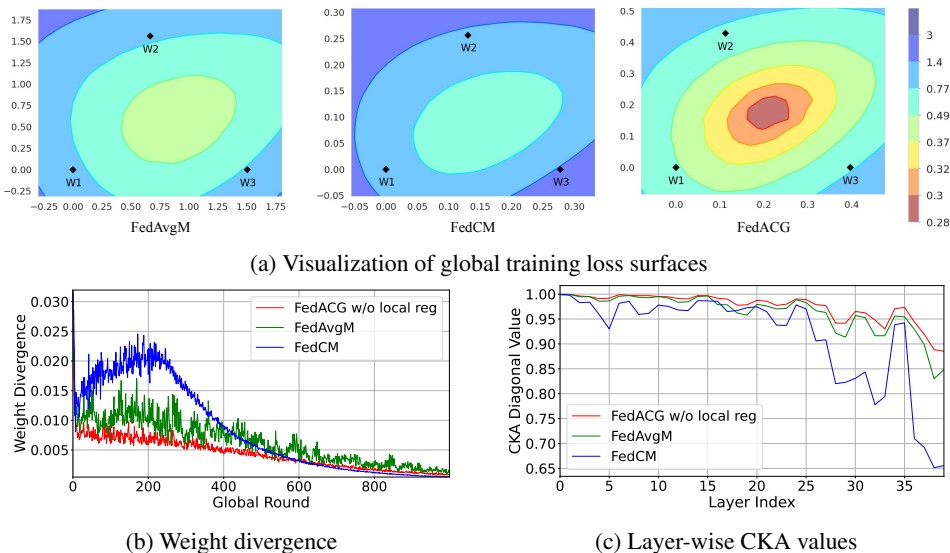

(a) Visualization of global training loss surfaces

(b) Weight divergence

(c) Layer-wise CKA values

Figure 2: **Benefit of accelerated client gradient.** For FedAvgM, FedCM, and FedACG (without local regularization for fair comparisons) on CIFAR10, we visualize (a) global training loss surfaces with three local models as black circles in the parameter space, (b) weight divergence, and (c) layer-wise CKA values. In (c), the $x$-axis denotes the layer index of ResNet-18 while the $y$-axis corresponds to CKA values measured on the global validation set.

and 50, respectively, each client has little training opportunity with few training examples and client heterogeneity increases significantly. As shown in Table 2, FedACG outperforms other methods for most cases, with the performance gap between FedACG and the second-best method, FedDC, being even more significant than when the participation rate is 2%, from -0.69%p to 1.39%p on CIFAR-10 and from 2.47%p to 6.34%p on CIFAR-100 at round 1000. This is partly because the local states in FedDC become stale quickly in this scenario, requiring extra iterations for convergence, whereas FedACG is not affected by this issue.

**Evaluation on dynamic client set** Since FedACG does not require storing local model history for local updates, it is conceptually better suited for scenarios with newly participating clients. To validate this property, we conduct an experiment, where we maintain 250 clients in each round but replace half of the clients on average every 100 rounds by setting the replacement probability of each client to 0.5. The experiment has been conducted on CIFAR-100 with Dirichlet(0.3) splits, assuming a participation ratio of 4% for each communication round. Table 3 shows that FedACG outperforms FedAvg and FedDyn. Note that FedDyn performs worse than FedAvg since the client models suffers from heterogeneity and divergence when new clients have no informative local states.

## 4.3 ANALYSIS

**Ablation study** Table 4 presents the contributions of individual components in the experiment on CIFAR-10 for the large-scale setting. We observe that the accelerated client gradient for local training makes a more critical impact on accuracy after 1,000 rounds. It is worth noting that the proposed regularization term in the local loss function shows a larger performance gain when used with the accelerated client gradient while employing the regularization term alone does not necessarily lead to a beneficial outcome on CIFAR-10 and CIFAR-100.

**Comparison with FedAvgM and FedCM** To better understand the effectiveness of the accelerated client gradient, we compare two momentum-based algorithms, FedAvgM and FedCM, by visualizing global loss surfaces, weight divergence, and layer-wise CKA values during training. Figure 2a highlights a better generalization of FedACG's local models to global loss compared to other methods. Figures 2b and 2c reveal that the local models of FedACG exhibit smaller divergence in the parameter space and more consistent feature representations, respectively. These findings demonstrate that the accelerated client gradient in FedACG effectively mitigates client drift stemming from data heterogeneity.

Table 5: Ablation study for $\lambda$ (a) and $\beta$ (b), w.r.t the accuracy at $1K^{\text{th}}$ round on CIFAR-10.

(a) Sensitivity of FedACG with respect to $\lambda$

| $\lambda$ | 0.75 | 0.8 | 0.85 | 0.9 |
|---|---|---|---|---|
| Dir(0.3) | 81.32 | 82.52 | 82.80 | 82.64 |
| *i.i.d.* | 85.52 | 86.82 | 86.83 | 86.16 |

(b) Sensitivity of FedACG with respect to $\beta$

| $\beta$ | 0.001 | 0.01 | 0.1 | 1 |
|---|---|---|---|---|
| Dir(0.3) | 82.10 | 82.80 | 82.32 | 82.44 |
| *i.i.d.* | 86.54 | 86.83 | 86.72 | 85.92 |

Table 6: Results on the realistic datasets involving feature skewness and data imbalance between clients and the dataset in the language domain.

| Method | FEMNIST | | CelebA | | ShakeSpeare | | | |
|---|---|---|---|---|---|---|---|---|
| | Acc. (%, ↑) 500R | Rounds (↓) 78% | Acc. (%, ↑) 500R | Rounds (↓) 88% | Acc. (%, ↑) 500R | 1000R | Rounds (↓) 42% | 45% |
| FedAvg | 78.38 | 328 | 89.92 | 134 | 45.01 | 46.55 | 94 | 500 |
| FedProx | 78.34 | 328 | 89.90 | 132 | 45.09 | 46.29 | 99 | 477 |
| FedAvgM | 78.37 | 256 | 89.85 | 113 | 44.63 | 45.91 | 63 | 690 |
| FedADAM | 75.96 | 500+ | 87.00 | 500+ | 44.89 | 44.30 | 68 | 1000+ |
| FedDyn | 79.80 | 227 | 89.74 | 126 | 39.23 | 44.10 | 749 | 1000+ |
| MOON | 78.33 | 336 | 87.95 | 500+ | 42.02 | 42.65 | 499 | 1000+ |
| FedCM[†] | 72.79 | 500+ | 88.89 | 222 | - | - | - | - |
| FedDC[‡] | 80.11 | **149** | 88.97 | 126 | 30.62 | 44.27 | 926 | 1000+ |
| FedACG (ours) | **80.61** | 169 | **90.09** | **108** | **46.36** | **48.23** | **57** | **290** |

**Hyperparameters**  We test the accuracy of our algorithm for the Dirichlet(0.3) and *i.i.d.* splits by varying the values of $\lambda$ and $\beta$, which control the momentum integration of the server model and the weight of the proximal term, respectively. As shown in Table 5a, the performance of FedACG remains stable in a range of $\lambda$ values from 0.75 to 0.9. Despite minor fluctuations, the accuracy remains high, peaking at $\lambda = 0.85$. Table 5a also shows that the accuracy is stable with respect to $\beta$.

## 4.4 Experiments on realistic datasets

We conduct experiments on additional realistic datasets, FEMNIST and CelebA in LEAF (Caldas et al., 2019), which include other non-*i.i.d.* scenarios such as feature skewness and data imbalance between clients. For these experiments, the number of clients is set to 2,000 with data splits following Caldas et al. (2019) and 10 randomly sampled clients participate in training for each communication round. We employ a two-layer CNN for FEMNIST and a four-layer CNN for CelebA. Table 6 illustrates that FedACG also outperforms other baselines on both datasets for most cases, which highlighting its robustness to heterogeneity with data quantity and feature alignment. Note that, while FedACG requires 20 more communication rounds than FedDC to reach the target accuracy on FEMNIST, it sends 56.8% fewer parameters than FedDC.

We also evaluate FedACG in a different domain, next word prediction task, on the ShakeSpeare in LEAF, which also involves a significant data imbalance between clients. We adopt an LSTM as the backbone network and the client participation rate per round is set to 5%. Table 6 presents that FedACG is also effective for the language domain while FedCM exhibits poor performance even with extensive hyperparameter tuning.

## 5 Conclusion

This paper addresses a realistic federated learning scenario, where a large number of clients with heterogeneous data and limited participation constraints hinder the convergence and performance of trained models. To tackle these issues, we proposed a novel federated learning framework that aggregates past global gradient information for guiding client updates and regularizes the local update directions aligned with the global information. The proposed algorithm provides global gradient information to individual clients without incurring additional communication or memory overhead. We provided the proof for the convergence rate of the proposed approach. We demonstrated the effectiveness of the proposed algorithm in terms of robustness and communication efficiency in the presence of client heterogeneity through extensive evaluation on multiple benchmarks.

**Ethics statement**    We introduce a communication-efficient federated learning framework designed to manage non-*i.i.d* data distribution across remote clients. While our method inherently provides a basic level of privacy by not accessing the raw data stored on remote devices, there may be the potential threat of privacy breaches from malicious users . Therefore, when combined with appropriate defense strategies, we believe that our framework will further enhance privacy security, raising the overall privacy assurance.

**Reproducibility statement**    We present the procedure of our proposed method in Algorithm 1. We also present the implementation details and algorithm-dependent hyperparameters in Appendix A.1. We have submitted the code and will make it publicly available.

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

# A  EXPERIMENTS

In this section, we provide the detailed experimental setups used in our study. This outlines the evaluation metrics, implementation details, choices of hyperparameters, and additional experiments.

## A.1  SETUPS

**Evaluation metrics**  To evaluate the generalization performance of the methods, we use the entire test set of CIFAR-10, CIFAR-100, and Tiny-ImageNet. Since both the training speed as well as the final accuracy are important factors in federated learning, we measure: (i) the performance achieved at a specified number of rounds and (ii) the number of rounds required for an algorithm to attain the desired level of target accuracy, following Al-Shedivat et al. (2021). For the target accuracies, we first select the median of all the compared methods at round 1000 and another representative value lower than the median. For the methods that could not achieve the target accuracies within the maximum communication round, we append a + sign to the communication round number.

**Implementation details**  We adopt PyTorch (Paszke et al., 2019) to implement FedACG and the other baselines. We follow the evaluation protocol of Acar et al. (2021) and Xu et al. (2021). For local updates, we use the SGD optimizer with a learning rate of 0.1 for all approaches on the three benchmarks. We apply no momentum to the local SGD, but incorporate the weight decay of 0.001 to prevent overfitting. We also employ gradient clipping to increase the stability of the algorithms.

For the experiments on CIFAR-10 and CIFAR-100, we choose 5 as the number of local training epochs (50 iterations) and 0.1 as the local learning rate. We set the batch size of the local update to 50 and 10 for the 100 and 500 client participation, respectively. The learning rate decay parameter of each algorithm is selected from {0.995, 0.998, 1} to achieve the best performance. The global learning rate is set to 1, except for FedAdam, which is tested with 0.01.

For the experiments on Tiny-ImageNet, we match the total local iterations of local updates with other benchmarks by setting the batch size of local updates as 100 and 20 for the 100 and 500 client participation, respectively.

**Hyperparameter selection**  For the hyperparameter selection, we assume the scenario that the server can compute the validation accuracy through communication with clients at the early stages, which is common to all algorithms. $\alpha$ in FedCM is selected from {0.1, 0.3, 0.5}, $\alpha$ in FedDyn is selected from {0.001, 0.01, 0.1}, and $\alpha$ in FedDC is set to 0.01. $\tau$ in FedAdam is set to 0.001 while $\mu$ in MOON is set to 1. $\beta$ in FedAvgM is selected from {0.4, 0.6, 0.8}, $\beta$ in FedProx and FedACG is selected from {0.1, 0.01, 0.001}, and $\lambda$ in FedACG is selected from {0.8, 0.85, 0.9}.

## A.2  ADDITIONAL ANALYSIS FOR THE EFFECT OF ACCELERATED CLIENT GRADIENT

FedACG uses a lookahead model, $\theta^{t-1} + \lambda m^{t-1}$, to start local training. This helps clients match their local solutions with the global loss, ensuring consistent updates. We observe more empirical evidence that supports our claim.

Figure 3 shows the convergence curves of FedACG and FedAvgM on CIFAR-10 in the moderate-scale setting without smoothing. For the experiments, we set the momentum coefficient to 0.85 for both algorithms. We observe that FedACG consistently outperforms FedAvgM and has a smaller accuracy variation throughout the training procedure. Specifically, when we compute the average squared difference between the accuracy at time step $t$ without smoothing ($\text{Acc}^t$) and the accuracy given by the simple moving average ($\text{Acc}^t_{\text{SMA}}$) over 1,000 rounds of communication, *i.e.*, $\frac{1}{T} \sum_{t=0}^{T-1} (\text{Acc}^t - \text{Acc}^t_{\text{SMA}})^2$, the differences are 2.26 and 10.30 for FedACG and FedAvgM, respectively. We believe that this is partly because the proposed accelerated gradient allows each client's update to compensate for the potential noise in momentum, which is possible because the local updates start from the anticipated point, $\theta^{t-1} + \lambda m^{t-1}$.

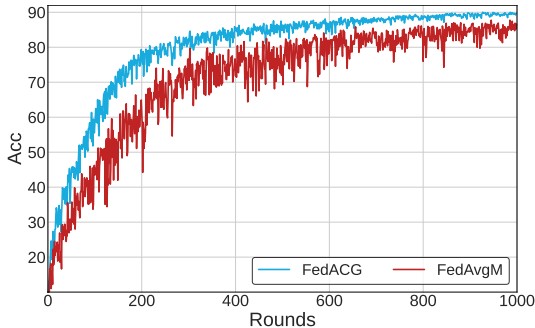

Figure 3: Training curves of FedACG and FedAvgM on CIFAR-10 in a moderate-scale setting without smoothing (exponential moving average).

## A.3 FEDACG WITH OTHER LOCAL OBJECTIVES

In Table 7, we incorporate accelerated client gradient into a client-side optimization technique, FedMLB (Kim et al., 2022), to test its benefits. "+ACG" means adopting the relaxed initialization. It shows that the relaxed initialization helps client-side optimization approaches achieve significant improvements without any additional communication costs.

Table 7: We incorporated accelerated client gradient (ACG) into client-side optimization technique on CIFAR-100 and Tiny-ImageNet.

(a) 100 clients, 5% participation, Dirichlet (0.3)

| Method | CIFAR-100 | | | | Tiny-ImageNet | | | |
|---|---|---|---|---|---|---|---|---|
| | Acc. (%, ↑) | | Rounds (↓) | | Acc. (%, ↑) | | Rounds (↓) | |
| | 500R | 1000R | 47% | 55% | 500R | 1000R | 35% | 38% |
| FedMLB (Kim et al., 2022) | 47.39 | 54.58 | 488 | 1000+ | 37.20 | 40.16 | 414 | 539 |
| FedMLB + ACG | **61.32** | **65.67** | **216** | **316** | **46.11** | **50.54** | **205** | **260** |

(b) 500 clients, 2% participation, Dirichlet (0.3)

| Method | CIFAR-100 | | | | Tiny-ImageNet | | | |
|---|---|---|---|---|---|---|---|---|
| | Acc. (%, ↑) | | Rounds (↓) | | Acc. (%, ↑) | | Rounds (↓) | |
| | 500R | 1000R | 36% | 40% | 500R | 1000R | 24% | 30% |
| FedMLB (Kim et al., 2022) | 32.30 | 42.61 | 642 | 800 | 28.39 | 33.67 | 384 | 489 |
| FedMLB + ACG | **41.10** | **55.27** | **402** | **479** | **35.92** | **43.85** | **209** | **313** |

## A.4 EVALUATION ON VARIOUS DATA HETEROGENEITY

Tables 8 and 9 show that FedACG matches or outperforms the performance of competitive methods when data heterogeneity is not severe (Dirichlet 0.6) or very low (*i.i.d.*) on CIFAR-10 in most cases. Note that, while the compared methods show performance degradation as the participation rate decreases, FedACG shows little degradation as the participation rate decreases for both data splits. This implies that FedACG is more robust for low participation rates than other baselines. This is partly because low client heterogeneity reduces noise in the momentum of the global gradient, which attributes to the smooth trajectory of global updates. Since FedACG effectively incorporates the momentum for local updates, FedACG is relatively unaffected by the partial participation of federated learning.

Table 8: Results with Dirichlet (0.6) data on CIFAR-10 and CIFAR-100 for two different settings.

(a) Dirichlet (0.6), 100 clients, 5% participation

| Method | CIFAR-10 | | | | CIFAR-100 | | | |
| --- | --- | --- | --- | --- | --- | --- | --- | --- |
| | Acc. (%, ↑) | | Rounds (↓) | | Acc. (%, ↑) | | Rounds (↓) | |
| | 500R | 1000R | 81% | 87% | 500R | 1000R | 50% | 56% |
| FedAvg (McMahan et al., 2017) | 80.56 | 85.97 | 520 | 1000+ | 43.91 | 49.18 | 1000+ | 1000+ |
| FedProx (Li et al., 2020a) | 80.39 | 85.53 | 524 | 1000+ | 43.15 | 48.45 | 1000+ | 1000+ |
| FedAvgM (Hsu et al., 2019) | 84.65 | 87.96 | 355 | 811 | 46.66 | 52.49 | 735 | 1000+ |
| FedADAM (Reddi et al., 2021) | 80.25 | 83.52 | 526 | 1000+ | 45.95 | 51.63 | 778 | 1000+ |
| FedDyn (Acar et al., 2021) | 87.23 | 89.49 | 310 | 487 | 50.51 | 56.78 | 488 | 886 |
| MOON (Li et al., 2021) | 84.95 | 87.99 | 272 | 728 | 55.76 | 61.42 | 338 | 527 |
| FedCM[†] (Xu et al., 2021) | 82.84 | 86.64 | 385 | 1000+ | 53.75 | 60.48 | 331 | 468 |
| FedDC[‡] (Gao et al., 2022) | **88.05** | 89.58 | 270 | **437** | 56.00 | 60.58 | 347 | 491 |
| FedACG (ours) | 87.57 | **90.56** | **218** | 453 | **58.82** | **63.88** | **243** | **396** |

(b) Dirichlet (0.6), 500 clients, 2% participation

| Method | CIFAR-10 | | | | CIFAR-100 | | | |
| --- | --- | --- | --- | --- | --- | --- | --- | --- |
| | acc. (%, ↑) | | rounds (↓) | | acc. (%, ↑) | | rounds (↓) | |
| | 500R | 1000R | 69% | 80% | 500R | 1000R | 32% | 41% |
| FedAvg (McMahan et al., 2017) | 62.79 | 75.17 | 671 | 1000+ | 29.41 | 36.62 | 648 | 1000+ |
| FedProx (Li et al., 2020a) | 62.48 | 75.10 | 688 | 1000+ | 29.62 | 36.70 | 647 | 1000+ |
| FedAvgM (Hsu et al., 2019) | 69.10 | 80.26 | 498 | 981 | 32.78 | 41.93 | 468 | 942 |
| FedADAM (Reddi et al., 2021) | 68.48 | 78.92 | 535 | 1000+ | 37.57 | 48.29 | 341 | 624 |
| FedDyn (Acar et al., 2021) | 68.53 | 80.33 | 513 | 983 | 32.06 | 43.28 | 498 | 917 |
| MOON (Li et al., 2021) | 74.29 | 80.66 | 368 | 921 | 31.64 | 41.61 | 515 | 931 |
| FedCM[†] (Xu et al., 2021) | 71.42 | 78.94 | 429 | 1000+ | 26.82 | 39.78 | 714 | 1000+ |
| FedDC[‡] (Gao et al., 2022) | 77.74 | **86.32** | 324 | 596 | 34.24 | 44.69 | 444 | 825 |
| FedACG (ours) | **78.49** | 85.28 | **289** | **565** | **39.61** | **49.70** | **304** | **540** |

Table 9: Results with *i.i.d.* data on CIFAR-10 and CIFAR-100 for two different settings.

(a) *i.i.d.*, 100 clients, 5% participation

| Method | CIFAR-10 | | | | CIFAR-100 | | | |
| --- | --- | --- | --- | --- | --- | --- | --- | --- |
| | acc. (%, ↑) | | rounds (↓) | | acc. (%, ↑) | | rounds (↓) | |
| | 500R | 1000R | 82% | 89% | 500R | 1000R | 52% | 58% |
| FedAvg (McMahan et al., 2017) | 85.28 | 88.69 | 372 | 1000+ | 43.96 | 48.20 | 1000+ | 1000+ |
| FedProx (Li et al., 2020a) | 84.79 | 87.99 | 384 | 1000+ | 43.57 | 47.75 | 1000+ | 1000+ |
| FedAvgM (Hsu et al., 2019) | 87.67 | 89.96 | 258 | 375 | 47.43 | 52.83 | 880 | 1000+ |
| FedADAM (Reddi et al., 2021) | 85.29 | 87.97 | 286 | 1000+ | 52.23 | 57.73 | 496 | 1000+ |
| FedDyn (Acar et al., 2021) | 89.19 | 90.70 | 269 | 492 | 50.37 | 56.88 | 592 | 898 |
| MOON (Li et al., 2021) | 88.24 | 89.96 | 207 | 628 | 58.50 | **64.73** | 311 | 484 |
| FedCM[†] (Xu et al., 2021) | 87.38 | 89.65 | 182 | 782 | 57.10 | 62.48 | 266 | 466 |
| FedDC[‡] (Gao et al., 2022) | 90.07 | 90.80 | 194 | 425 | 55.17 | 61.00 | 400 | 633 |
| FedACG (ours) | **90.57** | **92.29** | **157** | **354** | **59.82** | 64.08 | **244** | **342** |

(b) *i.i.d.*, 500 clients, 2% participation

| Method | CIFAR-10 | | | | CIFAR-100 | | | |
| --- | --- | --- | --- | --- | --- | --- | --- | --- |
| | acc. (%, ↑) | | rounds (↓) | | acc. (%, ↑) | | rounds (↓) | |
| | 500R | 1000R | 75% | 83% | 500R | 1000R | 33% | 42% |
| FedAvg (McMahan et al., 2017) | 68.70 | 78.21 | 652 | 1000+ | 30.71 | 37.85 | 664 | 1000+ |
| FedProx (Li et al., 2020a) | 68.74 | 77.96 | 643 | 1000+ | 30.11 | 37.13 | 685 | 1000+ |
| FedAvgM (Hsu et al., 2019) | 74.34 | 83.64 | 523 | 943 | 33.54 | 42.55 | 479 | 971 |
| FedADAM (Reddi et al., 2021) | 75.32 | 84.01 | 491 | 915 | 38.74 | 48.94 | 328 | 636 |
| FedDyn (Acar et al., 2021) | 74.81 | 84.71 | 398 | 823 | 33.20 | 42.91 | 492 | 936 |
| MOON (Li et al., 2021) | 69.86 | 81.89 | 586 | 1000+ | 28.82 | 41.26 | 649 | 1000+ |
| FedCM[†] (Xu et al., 2021) | 77.84 | 83.26 | 491 | 959 | 29.59 | 42.04 | 653 | 991 |
| FedDC[‡] (Gao et al., 2022) | **80.87** | **87.53** | 358 | **574** | 33.93 | 45.80 | 476 | 817 |
| FedACG (ours) | 80.15 | 87.47 | **316** | 578 | **41.16** | **49.10** | **299** | **525** |

## A.5 CONVERGENCE PLOT

### A.5.1 EVALUATION ON VARIOUS FEDERATED LEARNING SCENARIOS

Figure 4 to Figure 6 show the convergence of FedACG and the compared algorithms on CIFAR-10, CIFAR-100, and Tiny-ImageNet for various federated learning settings: varying the number of total clients, participation rates, data heterogeneity. FedACG continuously matches or exceeds the performance of the most powerful of our competitors in most learning sections.

Figure 7 shows the convergence plots under massive clients, 1% participation rate setting. The result shows that FedACG takes the lead in most learning sections, which also demonstrates the effectiveness of FedACG.

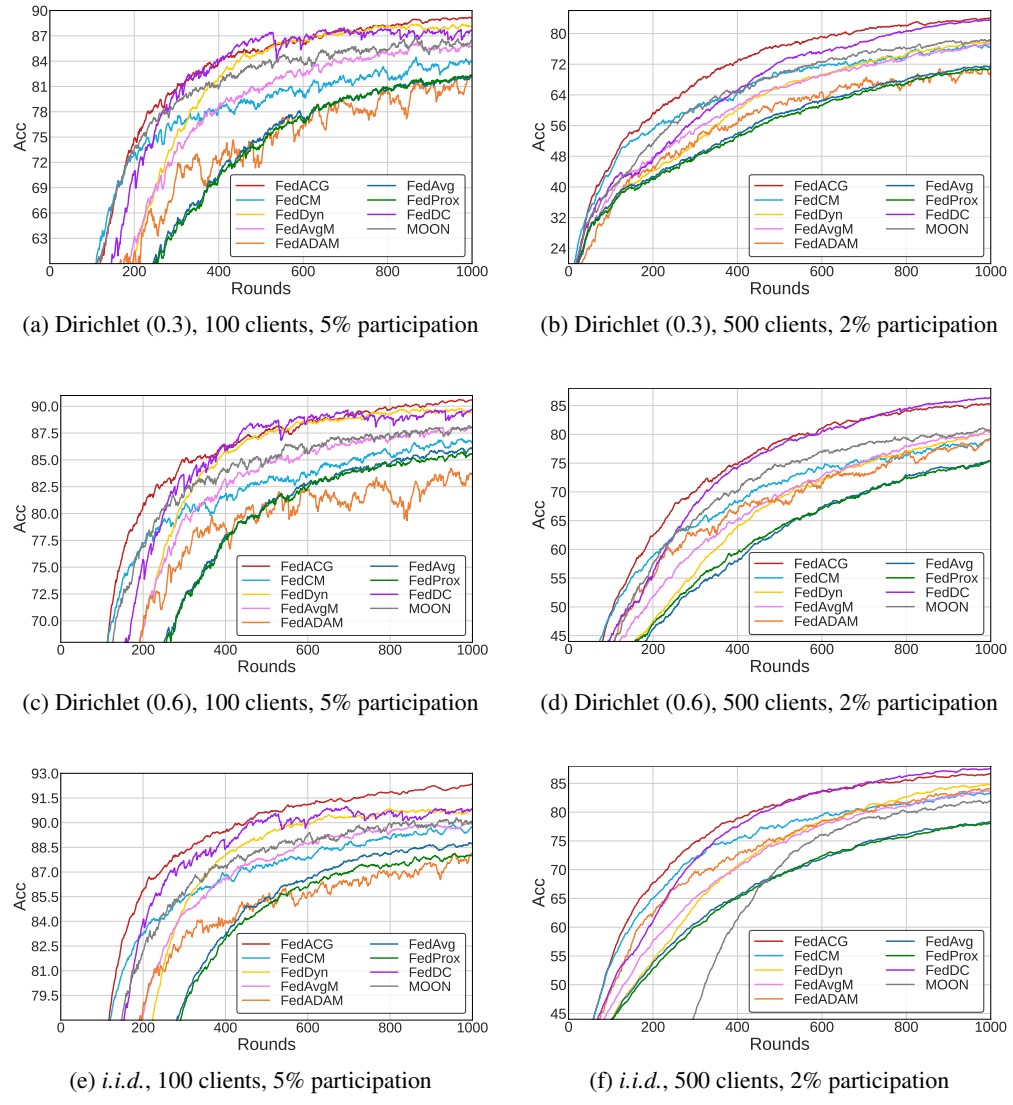

(a) Dirichlet (0.3), 100 clients, 5% participation  (b) Dirichlet (0.3), 500 clients, 2% participation

(c) Dirichlet (0.6), 100 clients, 5% participation  (d) Dirichlet (0.6), 500 clients, 2% participation

(e) *i.i.d.*, 100 clients, 5% participation  (f) *i.i.d.*, 500 clients, 2% participation

Figure 4: The convergence plots of FedACG and the baselines on CIFAR-10 with different federated learning scenarios.

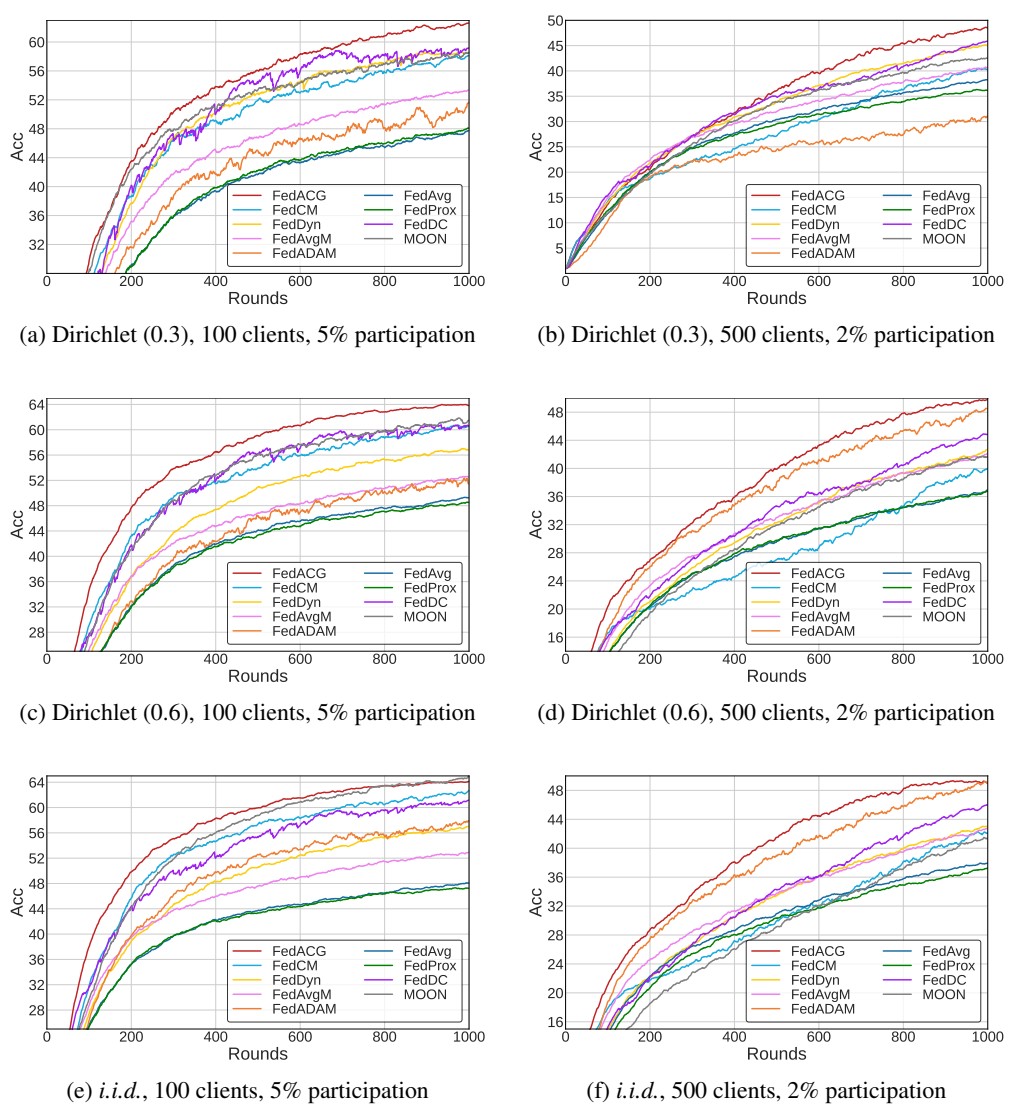

(a) Dirichlet (0.3), 100 clients, 5% participation     (b) Dirichlet (0.3), 500 clients, 2% participation

(c) Dirichlet (0.6), 100 clients, 5% participation     (d) Dirichlet (0.6), 500 clients, 2% participation

(e) *i.i.d.*, 100 clients, 5% participation     (f) *i.i.d.*, 500 clients, 2% participation

Figure 5: The convergence plots of FedACG and the baselines on CIFAR-100 with different federated learning scenarios.

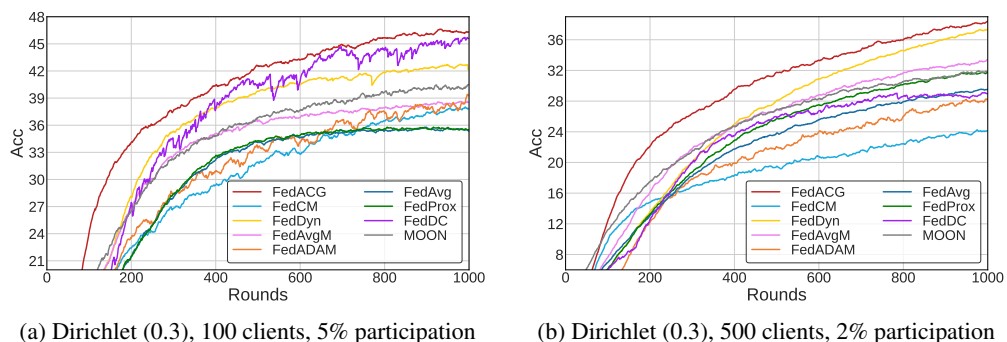

(a) Dirichlet (0.3), 100 clients, 5% participation     (b) Dirichlet (0.3), 500 clients, 2% participation

Figure 6: The convergence plots of FedACG and the baselines on Tiny-ImageNet with different federated learning scenarios.

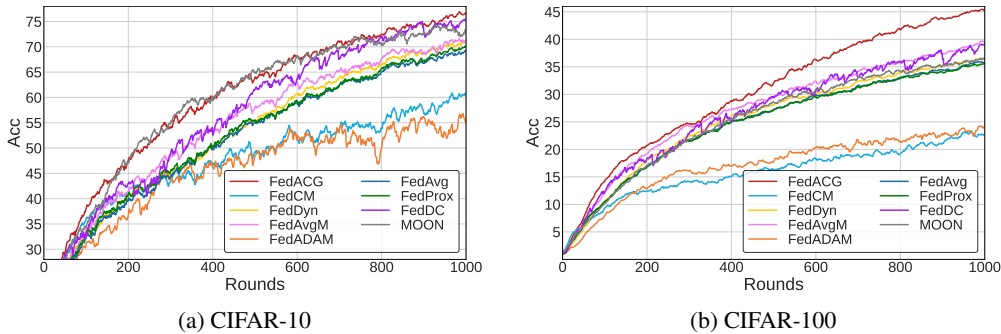

(a) CIFAR-10          (b) CIFAR-100

Figure 7: The convergence plots of FedACG and the baselines when the participation rate is low (1%) for 500 clients on CIFAR-10 and CIFAR-100. The Dirichlet parameter is set to 0.3 for the experiments.

### A.5.2 EVALUATION ON DYNAMIC CLIENT SET

Figure 8 shows a convergence plot when the entire client's pool changes during training. The result shows that FedACG outperforms the baselines in most learning sections. Note that FedDyn shows worse performance than FedAvg in the overall section of learning, and only achieves FedAvg's performance at the end. This is partly because it needs to store local states for local training in each client, which requires a kind of warm-up period for newly participating clients to contain useful information. In contrast, FedACG, which is free from these restrictions, shows strength in a realistic federated learning scenario where the pool of entire clients changes during training.

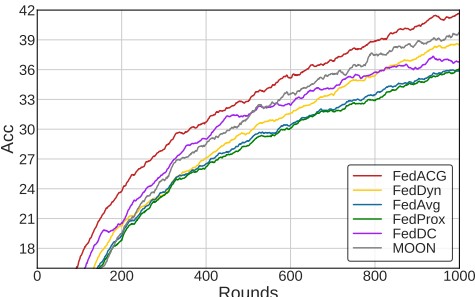

Figure 8: The convergence plots of FedACG, FedAvg, FedDyn, FedDC, and MOON on CIFAR-100 when the client set changes over dynamically: we sample 250 clients out of 500 clients as a candidate clients set at every 100 rounds over 10 stages on Dirichlet (0.3) split. 10 clients out of the sampled client set participate for the local training for each communication round. Dirichlet parameter is set to 0.3.

### A.6 INTEGRATION OF FEDACG WITH QUANTIZATION TECHNIQUES.

To validate that FedACG algorithm can be integrated into quantization-based techniques, we have conducted an experiment combining FedACG with FedPAQ (Reisizadeh et al., 2020), one of the quantization methods in federated learning, on the CIFAR-100, using 8 bits for quantization. Table 10 shows that our method works well even with quantization methods, surpassing both FedPAQ and FedAvg without quantization.

Table 10: Accuracy of FedACG with the integration of FedPAQ at the 1K round on CIFAR-100.

| Method | 100 clients (5% participation) | 500 clients (2% participation) |
| --- | --- | --- |
| FedAvg | 47.83 | 38.11 |
| FedPAQ | 41.99 | 30.21 |
| FedACG + FedPAQ | **50.53** | **39.55** |

## B  FEDACG VS. FEDAVGM AND FEDPROX

**FedAvgM (Hsu et al., 2019)**  The unique characteristic of FedACG lies in the initialization of client models. Although FedACG and FedAvgM use global momentum in common, FedACG broadcasts the accelerated global model by adding global momentum to the current global model $(\theta^{t-1} + \lambda m^{t-1})$ while FedAvgM broadcasts the current global model $(\theta^{t-1})$ to each client as the initial point of the local model. Figure 9 illustrates the difference in the server broadcasting and aggregation process of FedACG and FedAvgM, respectively. Unlike FedAvgM, where the only server knows the global momentum, in FedACG, the clients, the actual learners, start their local training at the anticipated point $\theta^{t-1} + \lambda m^{t-1}$ and can make up for the previous global momentum $\lambda m^{t-1}$ through their local training. This difference in the initial point seems to allow FedACG to be trained more responsive way, letting it behave more stably than FedAvgM in many federated learning settings.

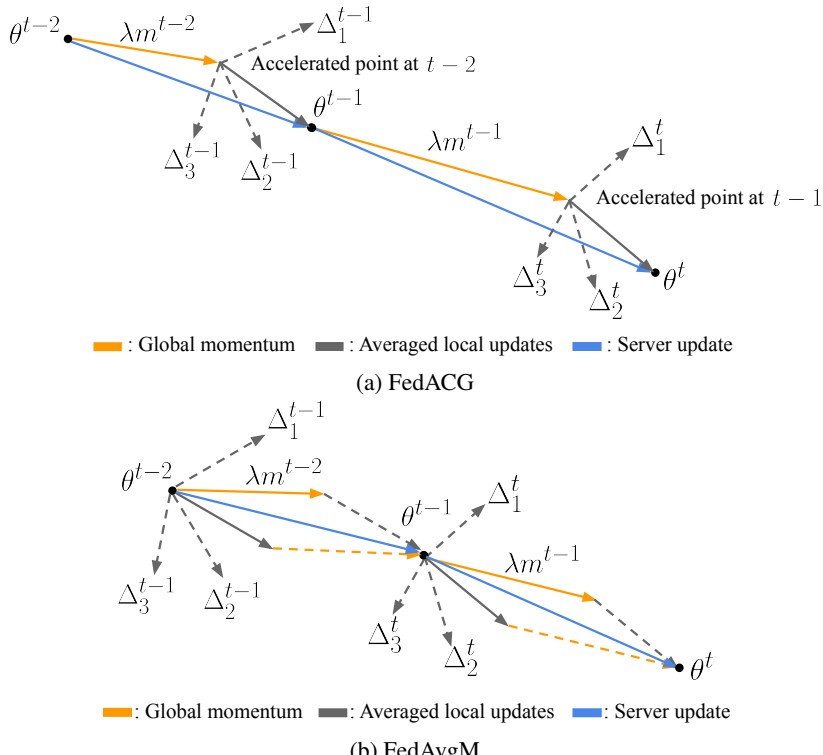

Figure 9: An illustration of the FedACG and FedAvgM during two communication rounds.

**FedProx (Li et al., 2020a)**  FedACG is a different method from FedProx for three principal reasons. First, FedACG utilizes the global momentum for server updates. Second, since FedACG uses client accelerated gradient, the local model's initial point is different from FedProx. From this, the third, objective function of FedACG regularizes the distance not between the local model and the previous global model (FedProx), but between the local model and the accelerated point.

## C  CONVERGENCE OF FEDACG

We now present the theoretical convergence rate of FedACG. We first state a few assumptions for the local loss functions $\mathcal{F}_i(\theta)$, which are commonly used in several previous works on federated optimization (Karimireddy et al., 2020; Reddi et al., 2021; Xu et al., 2021). First, the local function $\mathcal{F}_i(\cdot)$ is assumed to be $L$-smooth for all $i \in \{1, \ldots, N\}$, *i.e.*,

$$\|\nabla \mathcal{F}_i(x) - \nabla \mathcal{F}_i(y)\| \leq L\|x - y\| \ \ \forall x, y. \tag{3}$$

if local functions $\{\mathcal{F}_i(\cdot)\}_{i=1}^N$ are convex, we additionally have

$$\forall x, \qquad \frac{1}{2LN}\sum_{i=1}^N \|\nabla\mathcal{F}_i(x) - \nabla\mathcal{F}_i(x_*)\|^2 \leq \mathcal{F}(x) - \mathcal{F}(x_*) \text{ and} \qquad (4)$$

$$\forall x, y, z, \ \langle\nabla\mathcal{F}_i(x), z - y\rangle \leq -\mathcal{F}_i(z) + \mathcal{F}_i(y) + \frac{L}{2}\|z - x\|^2, \qquad (5)$$

where $\mathcal{F}(x) = \frac{1}{N}\sum_{i=1}^N \mathcal{F}_i(x)$ and $\nabla\mathcal{F}(x_*) = 0$. Second, we assume the stochastic gradient of the local loss function $\nabla f_i(x) := \nabla\mathcal{F}_i(x; \mathcal{D}_i)$ is unbiased and possesses a bounded variance, $i.e.$, $\mathbb{E}_{\mathcal{D}_i}\|\nabla f_i(x) - \nabla\mathcal{F}_i(x)\| < \sigma^2$. Third, we assume the average norm of local gradients is bounded by a function of the global gradient magnitude as $\frac{1}{N}\sum_{i=1}^N \|\nabla\mathcal{F}_i(x)\|^2 \leq \sigma_g^2 + B^2\|\nabla\mathcal{F}(x)\|^2$, where $\sigma_g \geq 0$ and $B \geq 1$. Based on the above assumptions, we derive the following asymptotic convergence bound of FedACG.

## C.1 Convergence of FedACG for non-convex functions

**Theorem 2.** *(Convergence of FedACG for non-convex functions) Suppose that local functions $\{\mathcal{F}_i\}_{i=1}^N$ are non-convex and $L$-smooth. Let $z^t = \theta^t + \frac{\lambda}{1-\lambda}m^t$. Then, by setting $\eta \leq \frac{1-\lambda}{100KL(B^2+1)}$, FedACG satisfies,*

$$\min_{t=1,\ldots,T}\mathbb{E}\left\|\nabla\mathcal{F}\left(\theta^{t-1} + \lambda m^{t-1}\right)\right\|^2 \leq 1000(B^2+1)\frac{Ld_0}{T} + \sqrt{\frac{40Ld_0}{T}\left(\psi_1\sigma^2 + \psi_2\sigma_g^2\right)}$$

$$+ \frac{1}{T^{\frac{2}{3}}}\left[800L^2d_0^2\ A\left(\psi_3\sigma^2 + \psi_4\sigma_g^2\right)\right]^{\frac{1}{3}}, \qquad (6)$$

*where $d_0 = \mathcal{F}(z^0) - \mathcal{F}(z^*)$, $A = 1 + \frac{4N}{(N-1)|S_t|}(1 - \frac{|S_t|}{N})$, $\psi_1 = A(\frac{1-\lambda}{K} + \frac{1}{K}) + \frac{5}{2} + (1-\lambda)(\frac{5}{K}+2)$, $\psi_2 = A((1-\lambda)^2 + 4) + 10(1-\lambda)$, $\psi_3 = \frac{1-\lambda}{20K}$, and $\psi_4 = \frac{(1-\lambda)^2}{10}$.*

*Proof.* We start the proof from the result in Lemma 3,

$$\mathbb{E}\left[\mathcal{F}\left(z^T\right) - \mathcal{F}\left(z^0\right)\right] \leq T\mathrm{C}_8\sigma^2 + T\mathrm{C}_9\sigma_g^2 + \mathrm{C}_{10}\sum_{t=1}^T \mathbb{E}\|\nabla\mathcal{F}\left(\theta_0^t\right)\|^2$$

where $\mathrm{C}_8 \leq \left(\frac{\eta k}{1-\lambda}\right)^2 L\left[(\frac{5}{2} + (1-\lambda)(\frac{5}{K} + 2)) + \frac{1}{K}\left\{(1-\lambda) + 1 + \frac{\eta KL}{20}\right\}A\right]$, $\mathrm{C}_9 \leq (\frac{\eta K}{1-\lambda})^2 L\left[10(1-\lambda) + \frac{\eta K}{1-\lambda}\frac{1-\lambda}{10}LA + A\left((1-\lambda^2) + 4\right)\right]$, and $\mathrm{C}_{10} \leq -\frac{\eta k}{10(1-\lambda)}$, respectively.

By rewriting the above inequalities,

$$\frac{1}{10}\frac{\eta K}{1-\lambda}\sum_{t=1}^T \mathbb{E}\|\nabla\mathcal{F}\left(\theta_0^t\right)\|^2 \leq -\mathrm{C}_{10}\sum_{t=1}^T \mathbb{E}\|\nabla\mathcal{F}\left(\theta_0^t\right)\|^2$$

$$\leq \mathbb{E}[\mathcal{F}(z^0) - \mathcal{F}(z^*)] + T\mathrm{C}_8\sigma^2 + T\mathrm{C}_9\sigma_g^2$$

Then

$$
\frac{1}{T}\sum_{t=1}^{T}\mathbb{E}\|\nabla\mathcal{F}\left(\theta_0^t\right)\|^2 \le \frac{1-\lambda}{\eta K}\frac{10}{T}\left(\mathcal{F}\left(z^0\right)-\mathcal{F}\left(z^*\right)\right) + \frac{1-\lambda}{\eta K}C_8\sigma^2 + \frac{1-\lambda}{\eta K}C_9\sigma_g^2
$$

$$
\le \frac{1-\lambda}{\eta}\frac{10}{T}\left(\mathcal{F}\left(z^0\right)-\mathcal{F}\left(z^*\right)\right) + \left(\frac{\eta K L}{1-\lambda}\right)\left[\left(\frac{5}{2}+(1-\lambda)\left(\frac{5}{K}+2\right)\right)\right.
$$
$$
\left. + \left(\frac{1+(1-\lambda)}{K}\right)A\right]\sigma^2 + \eta^2\left(\frac{KL}{1-\lambda}\right)^2\frac{1-\lambda}{20K}A\sigma^2
$$
$$
+ \frac{\eta K L}{1-\lambda}\left[10(1-\lambda)+A\left((1-\lambda)^2+4\right)\right]\sigma_g^2 + \frac{\eta^2 K^2 L^2}{10}A\sigma_g^2
$$
$$
\le \frac{10L}{T}(\frac{1-\lambda}{\eta K L})\left(\mathcal{F}\left(z^0\right)-\mathcal{F}\left(z^*\right)\right) + \frac{\eta K L}{1-\lambda}\left[\left\{A\left(\frac{(1-\lambda)}{K}+\frac{1}{K}\right)\right.\right.
$$
$$
\left. + \left(\frac{5}{2}+(1-\lambda)\left(\frac{5}{K}+2\right)\right)\right)\sigma^2 + \left\{A\left((1-\lambda)^2+4\right)+(10(1-\lambda))\right\}\sigma_g^2\right]
$$
$$
+ \left(\frac{\eta K L}{1-\lambda}\right)^2\left[A\left(\frac{1-\lambda}{20K}\sigma^2+\frac{(1-\lambda)^2}{10}\sigma_g^2\right)\right]
$$

Similar to the proof of Lemma 2 of the SCAFFOLD (Karimireddy et al., 2020), if we choose $\eta_{\max} \le \frac{1-\lambda}{100(B^2+1)KL}$, we can have

$$
\frac{1}{T}\sum_{t=1}^{T}\mathbb{E}\|\nabla\mathcal{F}\left(\theta_0^t\right)\|^2 \le \frac{1000\left(B^2+1\right)L}{T}\left(\mathcal{F}\left(z^0\right)-\mathcal{F}\left(z^*\right)\right)
$$
$$
+ \frac{2}{\sqrt{T}}\left[\left(10L(\mathcal{F}\left(z^0\right)-\mathcal{F}\left(z^*\right))\right)\left[\left\{A\left(\frac{2-\lambda}{K}\right)+\left(\frac{5}{2}+(1-\lambda)\left(\frac{5}{K}+2\right)\right)\right\}\sigma^2\right.\right.
$$
$$
+ \left\{A\left((1-\lambda)^2+4\right)+(10(1-\lambda))\right\}\delta_g^2\right]\right]^{\frac{1}{2}}
$$
$$
+ \frac{2}{T^{\frac{2}{3}}}\left(100L^2\left(\mathcal{F}\left(z^0\right)-\mathcal{F}\left(z^*\right)\right)^2\left[A\left(\frac{1-\lambda}{20K}\sigma^2+\frac{(1-\lambda)^2}{10}\sigma_g^2\right]\right]\right)^{\frac{1}{3}}
$$

We then complete the proof by noting that $\theta_0^t = \theta^{t-1}+\lambda m^{t-1}$. $\qquad\square$

**Lemma 3.** *For proving Theorem 2, we first prove the key Lemma below. Let $z^t = \theta^t + \frac{\lambda}{1-\lambda}m^t$, $\Delta_i^t = \theta_i^t - (\theta^{t-1}+\lambda m^{t-1}) = \sum_{k=0}^{K-1}-\eta\nabla f_i(\theta_{i,k}^t)$, $\delta^t = \frac{1}{N}\sum_{i\in[N]}\sum_{k=0}^{K-1}-\eta\nabla\mathcal{F}_i(\theta_{i,k}^t)$, $e^t = \Delta^t - \delta^t$, and $\theta_0^t = \theta^{t-1}+\lambda m^{t-1}$,*

$$
\mathbb{E}\left[\mathcal{F}\left(z^T\right)-\mathcal{F}\left(z^0\right)\right] \le TC_8\sigma^2 + TC_9\sigma_g^2 + C_{10}\sum_{t=1}^{T}\mathbb{E}\|\nabla\mathcal{F}\left(\theta_0^t\right)\|^2
$$

*where $C_8 \le \left(\frac{\eta k}{1-\lambda}\right)^2 L\left[\left(\frac{5}{2}+(1-\lambda)(\frac{5}{K}+2)\right)+\frac{1}{K}\left\{(1-\lambda)+1+\frac{\eta K L}{20}\right\}A\right]$, $C_9 \le \left(\frac{\eta K}{1-\lambda}\right)^2 L\left[10(1-\lambda)+\frac{\eta K}{1-\lambda}\frac{1-\lambda}{10}LA+A\left((1-\lambda^2)+4\right)\right]$, and $C_{10} \le -\frac{\eta k}{10(1-\lambda)}$, respectively.*

*Proof.*

$$
\begin{aligned}
\mathcal{F}(z^{t+1}) \leq\ & \mathcal{F}(z^t) + \langle \nabla \mathcal{F}(z^t), z^{t+1} - z^t \rangle + \frac{L}{2}\|z^{t+1} - z^t\|^2 \\
=\ & \mathcal{F}(z^t) + \frac{1}{1-\lambda}\langle \nabla \mathcal{F}(z^t), \Delta^{t+1} \rangle + \frac{L}{2(1-\lambda)^2}\|\Delta^{t+1}\|^2 \\
=\ & \mathcal{F}(z^t) + \frac{1}{1-\lambda}\langle \nabla \mathcal{F}(z^t), (e^{t+1} + \delta^{t+1}) + \eta K \nabla \mathcal{F}(\theta_0^{t+1}) - \eta K \nabla \mathcal{F}(\theta_0^{t+1}) \rangle \\
& + \frac{L}{2(1-\lambda)^2}\|\Delta^{t+1}\|^2 \\
=\ & \mathcal{F}(z^t) + \frac{1}{1-\lambda}\langle \nabla \mathcal{F}(z^t), e^{t+1} \rangle + \frac{1}{1-\lambda}\langle \nabla \mathcal{F}(z^t), \delta^{t+1} + \eta K \nabla \mathcal{F}(\theta_0^{t+1}) \rangle \\
& - \frac{\eta K}{1-\lambda}\langle \nabla \mathcal{F}(z^t), \nabla \mathcal{F}(\theta_0^{t+1}) \rangle + \frac{L}{2(1-\lambda)^2}\|\Delta^{t+1}\|^2 \\
=\ & \mathcal{F}(z^t) + \frac{1}{1-\lambda}\langle \nabla \mathcal{F}(z^t), e^{t+1} \rangle + \frac{1}{1-\lambda}\langle \nabla \mathcal{F}(z^t), \delta^{t+1} + \eta K \nabla \mathcal{F}(\theta_0^{t+1}) \rangle \\
& - \frac{\eta K}{1-\lambda}\langle \nabla \mathcal{F}(z^t) - \nabla \mathcal{F}(\theta_0^{t+1}), \nabla \mathcal{F}(\theta_0^{t+1}) \rangle - \frac{\eta K}{1-\lambda}\|\nabla \mathcal{F}(\theta_0^{t+1})\|^2 \\
& + \frac{L}{2(1-\lambda)^2}\|\Delta^{t+1}\|^2
\end{aligned}
$$

First inequality comes from the $L$-smoothness of the loss function $\mathcal{F}$. Note that $\theta_0^{t+1} = \theta^t + \lambda m^t$ represents the accelerated global model and serves as the initial point for the local models of the participating local clients at the round $t+1$, meaning that $\theta_0^{t+1} = \theta_{i,0}^{t+1}$ holds for each client $i \in S_t$. By taking expectation on both sides, we get following equation.

$$
\begin{aligned}
\mathbb{E}[\mathcal{F}(z^{t+1}) - \mathcal{F}(z^t)] \leq\ & \frac{1}{1-\lambda}\mathbb{E}\left[\langle \nabla \mathcal{F}\left(z^t\right), \delta^{t+1} + \eta K \nabla \mathcal{F}\left(\theta_0^{t+1}\right) \rangle\right] \\
& - \frac{\eta K}{1-\lambda}\mathbb{E}\left[\langle \nabla \mathcal{F}\left(z^t\right) - \nabla \mathcal{F}\left(\theta_0^{t+1}\right), \nabla \mathcal{F}\left(\theta_0^{t+1}\right) \rangle\right] \\
& - \frac{\eta K}{1-\lambda}\mathbb{E}\left\|\nabla \mathcal{F}\left(\theta_0^{t+1}\right)\right\|^2 + \frac{L}{2(1-\lambda)^2}\mathbb{E}\left\|\Delta^{t+1}\right\|^2 \\
\leq\ & \frac{K^2 L}{2(1-\lambda)} \underbrace{\mathbb{E}\left\|\eta \nabla \mathcal{F}\left(z^t\right)\right\|^2}_{\text{I}^*} + \frac{1}{2(1-\lambda)K^2 L}\underbrace{\mathbb{E}\|\frac{1}{\eta}\left(\delta^{t+1} + \eta K \nabla \mathcal{F}\left(\theta_0^{t+1}\right)\right)\|^2}_{\text{II}^*} \\
& + \frac{\eta K}{2(1-\lambda)}\underbrace{\mathbb{E}\left\|\nabla \mathcal{F}\left(z^t\right) - \nabla \mathcal{F}\left(\theta_0^{t+1}\right)\right\|^2}_{\text{III}^*} + \frac{\eta K}{1-\lambda}\left(\frac{1}{2} - 1\right)\mathbb{E}\left\|\nabla \mathcal{F}\left(\theta_0^{t+1}\right)\right\|^2 \\
& + \frac{L}{2(1-\lambda)^2}\underbrace{\mathbb{E}\left\|\Delta^{t+1}\right\|^2}_{\text{IV}^*}
\end{aligned}
$$

The first ineqaulity holds because $\mathbb{E}[e^{t+1}] = 0$. The second inequality comes from the Lemma 11. Now we have to find the upper bound of four terms depicted in the last inequality. To do so, we will first define a few terms for our analysis.

$$
C_i^t = \mathbb{E}\left\|\sum_{k=0}^{K-1} -\eta \nabla \mathcal{F}_i\left(\theta_{i,k}^t\right)\right\|^2, \quad C^t = \frac{1}{N}\sum_{i \in N} C_t^i
$$

$$
I_{i,k}^t = \mathbb{E}\|\theta_{i,k}^t - \theta_{i,0}^t\|^2, \quad I_i^t = \sum_{k=0}^{K-1} I_{i,k}^t, \quad I^t = \frac{1}{N}\sum_{i \in N} I_i^t
$$

$C_i^t$ is the expectation of the local updates for each client $i$ at $t^{\text{th}}$ global round, and $I_{i,k}^t$ is the expectation of the parameter difference between local model and the initial point of a local model at each $k^{\text{th}}$ local iteration. Using the above definition, the upper bound of $I^*$, $II^*$, $III^*$, and $IV^*$ are handled in Lemma 4, Lemma 5, Lemma 6, and Lemma 7, respectively. Substituting the upper bounds for the four items yields the following result.

$$
\begin{aligned}
\mathbb{E}[\mathcal{F}(z^{t+1}) - \mathcal{F}(z^t)] &\leq \frac{K^2 L}{2(1-\lambda)} \left\{ \frac{2\eta^2 L^2 \lambda^4}{(1-\lambda)^2} \mathbb{E}\left\|m^t\right\|^2 + 2\eta^2 \mathbb{E}\left\|\nabla \mathcal{F}\left(\theta_0^{t+1}\right)\right\|^2 \right\} + \frac{L}{2(1-\lambda)K} I^{t+1} \\
&\quad + \frac{\eta K}{2(1-\lambda)} \frac{\lambda^4 L^2}{(1-\lambda)^2} \mathbb{E}\left\|m^t\right\|^2 \\
&\quad - \frac{\eta K}{2(1-\lambda)} \mathbb{E}\left\|\nabla \mathcal{F}\left(\theta_0^{t+1}\right)\right\|^2 + \frac{L}{2(1-\lambda)^2} \eta^2 K^2 \sigma^2 + \frac{L}{2(1-\lambda)^2} A C^{t+1} \\
&= \frac{L^2 \lambda^4}{(1-\lambda)^2} \left\{ \frac{L\eta^2 K^2}{1-\lambda} + \frac{\eta K}{2(1-\lambda)} \right\} \mathbb{E}\left\|m^t\right\|^2 \\
&\quad + \frac{L}{2(1-\lambda)K} I^{t+1} + \frac{L}{2(1-\lambda)^2} \eta^2 K^2 \sigma^2 \\
&\quad + \frac{L}{2(1-\lambda)^2} A C^{t+1} + \frac{\eta K}{1-\lambda} \left(-\frac{1}{2} + \eta K L\right) \mathbb{E}\left\|\nabla \mathcal{F}\left(\theta_0^{t+1}\right)\right\|^2,
\end{aligned}
$$

where $A = 1 + \frac{4N}{(N-1)|S_t|}(1 - \frac{|S_t|}{N})$. By substituting the upper bound of $I^t$ and $C^t$ handled in Lemma 8 and Lemma 9,

$$
\mathbb{E}[\mathcal{F}(z^{t+1}) - \mathcal{F}(z^t)] \leq \mathrm{C}_4 \mathbb{E}\|m^t\|^2 + \mathrm{C}_5 \sigma^2 + \mathrm{C}_6 \sigma_g^2 + \mathrm{C}_7 \mathbb{E}\left\|\nabla \mathcal{F}\left(\theta_0^{t+1}\right)\right\|^2
$$

, where $\mathrm{C}_4 = \frac{L^2 \lambda^4}{(1-\lambda)^2} \left\{ \frac{L\eta^2 K^2}{1-\lambda} + \frac{\eta K}{2(1-\lambda)} \right\}$, $\mathrm{C}_5 = \frac{\eta^2 K^2 L}{2(1-\lambda)^2} + \frac{5\eta^2 KL}{(1-\lambda)} + \frac{10\eta^4 K^3 L^3}{(1-\lambda)^2} A$, $\mathrm{C}_6 = \frac{10\eta^2 K^2 L}{1-\lambda} + \frac{20\eta^4 K^4 L^3}{(1-\lambda)^2} A + \frac{\eta^2 K^2 L}{(1-\lambda)^2} A$, and $\mathrm{C}_7 = \left(\frac{10\eta^2 K^2 L}{1-\lambda} + \frac{20\eta^4 K^4 L^3}{(1-\lambda)^2} A + \frac{\eta^2 K^2 L}{(1-\lambda)^2} A\right) B^2 + \frac{\eta K}{1-\lambda} \left(-\frac{1}{2} + \eta K L\right)$, respectively. By summing the above inequalities for $t = 0, \ldots, T-1$,

$$
\mathbb{E}[\mathcal{F}\left(z^T\right) - \mathcal{F}\left(z^0\right)] = T \left(\mathrm{C}_5 \sigma^2 + \left(\mathrm{C}_6 \sigma_g^2\right) + \mathrm{C}_7 \mathbb{E}\sum_{t=1}^{T}\left\|\nabla \mathcal{F}\left(\theta_0^t\right)\right\|^2\right) + \mathrm{C}_4 \sum_{t=1}^{T} \mathbb{E}\left\|m^{t-1}\right\|^2,
$$

By substituting the upper bound of $\sum_{t=1}^{T} \mathbb{E}\left\|m^{t-1}\right\|^2$ handled in Lemma 10 and organizing the coefficients of each term, we can get following result.

$$
\mathbb{E}\left[\mathcal{F}\left(z^T\right) - \mathcal{F}\left(z^0\right)\right] \leq T\mathrm{C}_8 \sigma^2 + T\mathrm{C}_9 \sigma_g^2 + \mathrm{C}_{10} \sum_{t=1}^{T} \mathbb{E}\|\nabla \mathcal{F}\left(\theta_0^t\right)\|^2
$$

where

$$
\begin{aligned}
\mathrm{C}_8 &= \left(\frac{\eta K}{1-\lambda}\right)^2 \left(\frac{L}{2} + \frac{5L}{K}(1-\lambda) + \left(\frac{\eta K L}{1-\lambda}\right)^2 \frac{10L(1-\lambda)}{K} A\right) \\
&\quad + \left(\frac{\eta K}{1-\lambda}\right)^2 \frac{\lambda^4}{(1-\lambda)^4} L \left\{ nKL + \frac{1}{2} \right\} (\eta K L(1-\lambda) + 20\frac{\eta^3 K^3 L^3}{K} A)
\end{aligned}
$$

$$
\begin{aligned}
\mathrm{C}_9 &= \frac{10\eta^2 K^2 L}{(1-\lambda)} + \frac{20\eta^4 K^4 L^3}{(1-\lambda)^2} A + \frac{\eta^2 K^2 L}{(1-\lambda)^2} A \\
&\quad + \frac{L^2 \lambda^4}{(1-\lambda)^4} \left\{ \frac{L^2 \eta^2 K^2}{1-\lambda} + \frac{\eta K}{2(1-\lambda)} \right\} A(40\eta^4 K^4 L^2 + 2\eta^2 K^2)
\end{aligned}
$$

$$C_{10} = \left[\frac{10\eta^2 K^2 L}{(1-\lambda)} + \frac{20\eta^4 K^4 L^3}{(1-\lambda)^2}A + \frac{n^2 K^2 L}{(1-\lambda)^2}A\right]B^2 + \frac{\eta K}{1-\lambda}\left(-\frac{1}{2} + \eta KL\right)$$
$$+ \frac{L^2\lambda^4}{(1-\lambda)^4}\left[\left\{\frac{L\eta^2 K^2}{1-\lambda} + \frac{\eta K}{2(1-\lambda)}\right\}B^2 A(40\eta^4 K^4 L^2 + 2\eta^2 K^2)\right]$$

The optimal value of $\lambda$, which minimizes the upper bound of $\mathbb{E}\left[\mathcal{F}\left(z^T\right) - \mathcal{F}\left(z^0\right)\right]$, lies within the interval of 0 to 1. Within the specified range defined by $\lambda^2 \leq 25(1-\lambda) \Rightarrow \lambda \leq 0.962 \approx 1$, and with the appropriate selection of $\eta$, an increase in $\lambda$ results in a decrease in the upper bound of $\mathbb{E}\left[\mathcal{F}\left(z^T\right) - \mathcal{F}\left(z^0\right)\right]$. In the conditions described above, with respect to $\eta$ and $\lambda$, we have following upper bounds for $C_8$, $C_9$, and $C_{10}$,

$$C_8 \leq \left(\frac{\eta k}{1-\lambda}\right)^2 L\left[\left(\frac{5}{2} + (1-\lambda)(\frac{5}{K} + 2)\right) + \frac{1}{K}\left\{(1-\lambda) + 1 + \frac{\eta KL}{20}\right\}A\right]$$

$$C_9 \leq (\frac{\eta K}{1-\lambda})^2 L\left[10(1-\lambda) + \frac{\eta K}{1-\lambda}\frac{1-\lambda}{10}LA + A\left((1-\lambda^2) + 4\right)\right]$$

$$C_{10} \leq -\frac{\eta k}{10(1-\lambda)}$$

which gives us the desired result. $\qquad\square$

**Lemma 4.** $\mathbb{E}\|\eta\nabla\mathcal{F}\left(z^t\right)\|^2$ *in the proof of Lemma 3 has following bound.*

$$\mathbb{E}\|\eta\nabla\mathcal{F}\left(z^t\right)\|^2 \leq 2\eta^2 L^2\frac{\lambda^4}{1-\lambda}\mathbb{E}\|m^t\|^2 + 2\eta^2\mathbb{E}\|\nabla\mathcal{F}\left(\theta_0^{t+1}\right)\|^2$$

*Proof.*

$$\mathbb{E}\|\eta\nabla\mathcal{F}\left(z^t\right)\|^2 = \mathbb{E}\|\eta\nabla\mathcal{F}\left(z^t\right) - \eta\nabla\mathcal{F}\left(\theta_{i,0}^{t+1}\right) + \eta\nabla\mathcal{F}\left(\theta_{i,0}^{t+1}\right)\|^2$$
$$\leq 2\eta^2 L^2\mathbb{E}\|\frac{\lambda^2}{1-\lambda}m^t\|^2 + 2\eta^2\mathbb{E}\|\nabla\mathcal{F}\left(\theta_0^{t+1}\right)\|^2$$
$$\leq 2\eta^2 L^2\frac{\lambda^4}{(1-\lambda)^2}\mathbb{E}\|m^t\| + 2\eta^2\mathbb{E}\|\nabla\mathcal{F}\left(\theta_0^{t+1}\right)\|^2$$

$\qquad\square$

**Lemma 5.** $\mathbb{E}\|\frac{1}{\eta}(\delta^{t+1} + \eta K\nabla\mathcal{F}(\theta_0^{t+1}))\|^2$ *in the proof of Lemma 3 has following bound.*

$$\mathbb{E}\|\frac{1}{\eta}(\delta^{t+1} + \eta K\nabla\mathcal{F}(\theta_0^{t+1}))\|^2 \leq L^2 K I^{t+1}$$

*Proof.*

$$\mathbb{E}\|\frac{1}{\eta}(\delta^{t+1} + \eta K\nabla\mathcal{F}(\theta_0^{t+1}))\|^2 = \mathbb{E}[\|\frac{1}{\eta}(\delta^{t+1} + \eta K\nabla\mathcal{F}(\theta^t + \lambda m^t))\|^2$$
$$= \mathbb{E}\|\frac{1}{N}\sum_{i\in[N]}\sum_{k=0}^{K-1}\{-\nabla\mathcal{F}_i(\theta_{i,k}^{t+1}) + \nabla\mathcal{F}_i(\theta^t + \lambda m^t))\}\|^2$$
$$\leq \frac{K}{N}\sum_{i\in[N]}\sum_{k=0}^{K-1}\mathbb{E}\|\{-\nabla\mathcal{F}_i(\theta_{i,k}^{t+1}) + \nabla\mathcal{F}_i(\theta^t + \lambda m^t))\}\|^2$$
$$\leq \frac{L^2 K}{N}\sum_{i\in[N]}\sum_{k=0}^{K-1}\mathbb{E}\|\theta_{i,k}^{t+1} - \theta_{i,0}^{t+1}\|^2$$
$$= L^2 K I^{t+1}$$

Inequality in the third line comes from Jensen's inequality. Inequality in the fourth line is derived by the smoothness of the objective function. Equality in the last line use the definition of $I^t$.

$\qquad\square$

**Lemma 6.** $\mathbb{E}\|\nabla\mathcal{F}(z^t) - \nabla\mathcal{F}(\theta_0^{t+1})\|^2$ *in the proof of Lemma 3 has the following bound for any* $0 \leq \lambda < 1$,

$$\mathbb{E}\|\nabla\mathcal{F}(z^t) - \nabla\mathcal{F}(\theta_0^{t+1})\|^2 \leq \frac{\lambda^4 L^2}{(1-\lambda)^2}\mathbb{E}\|m^t\|^2$$

*Proof.*

$$\mathbb{E}\|\nabla\mathcal{F}(z^t) - \nabla\mathcal{F}(\theta^t + \lambda m^t)\|^2 \leq L^2\mathbb{E}\|\frac{\lambda^2}{1-\lambda}m^t\|^2 = \frac{\lambda^4 L^2}{(1-\lambda)^2}\mathbb{E}\|m^t\|^2$$

*where the inequality comes from the $L$-smoothness of the global loss function $\mathcal{F}$.* $\qquad\square$

**Lemma 7.** $\mathbb{E}\|\Delta^t\|^2$ *in the proof of Lemma 3 has the following bound,*

$$\mathbb{E}\|\Delta^t\|^2 \leq \eta^2 K^2 \sigma^2 + AC_t,$$

*where* $A = 1 + \frac{4N}{(N-1)|S_t|}(1 - \frac{|S_t|}{N})$

*Proof.* $\mathbb{E}\|\Delta^t\|^2 = \mathbb{E}\|\delta^t\|^2 + \mathbb{E}\|\Delta^t - \delta^t\|^2.$

For the first term $\mathbb{E}\|\delta^t\|^2$, we can derive upper bound as

$$\mathbb{E}\|\delta^t\|^2 = \mathbb{E}\|\frac{1}{N}\sum_{i\in[N]}\sum_{k=0}^{K-1} -\eta\nabla\mathcal{F}_i(\theta_{i,k}^t)\|^2 \leq \frac{1}{N}\sum_{i\in[N]}\mathbb{E}\|\sum_{k=0}^{K-1} -\eta\nabla\mathcal{F}_i(\theta_{i,k}^t)\|^2 = C^t,$$

where the inequality from Jensen's inequality.

For the second term $\mathbb{E}\|\Delta^t - \delta^t\|^2$,

$$\mathbb{E}\|\Delta^t - \delta^t\|^2 = \mathbb{E}\|\Delta^t + \frac{\eta}{|\mathcal{S}_t|}\sum_{i\in\mathcal{S}_t}\sum_{k=0}^{K-1}\nabla\mathcal{F}_i(\theta_{i,k}^t) - \frac{\eta}{|\mathcal{S}_t|}\sum_{i\in\mathcal{S}_t}\sum_{k=0}^{K-1}\nabla\mathcal{F}_i(\theta_{i,k}^t) + \frac{\eta}{N}\sum_{i\in[N]}\sum_{k=0}^{K-1}\nabla\mathcal{F}_i(\theta_{i,k}^t)\|^2$$

$$= \mathbb{E}\|\frac{\eta}{|\mathcal{S}_t|}\sum_{i\in\mathcal{S}_t}\sum_{k=0}^{K-1}\nabla f_i(\theta_{i,k}^t) - \nabla\mathcal{F}_i(\theta_{i,k}^t)\|^2 + \mathbb{E}\|\frac{\eta}{|\mathcal{S}_t|}\sum_{i\in\mathcal{S}_t}\sum_{k=0}^{K-1}\nabla\mathcal{F}_i(\theta_{i,k}^t) - \delta^t\|^2$$

$$\leq \frac{\eta^2}{|\mathcal{S}_t|}\sum_{i\in\mathcal{S}_t}\sum_{k=0}^{K-1}\mathbb{E}\|\nabla f_i(\theta_{i,k}^t) - \nabla\mathcal{F}_i(\theta_{i,k}^t)\|^2 + \mathbb{E}\|\frac{\eta}{|\mathcal{S}_t|}\sum_{i\in\mathcal{S}_t}\sum_{k=0}^{K-1}\nabla\mathcal{F}_i(\theta_{i,k}^t) - \delta^t\|^2$$

$$= \eta^2 K^2 \sigma^2 + \underbrace{\mathbb{E}\|\frac{\eta}{|\mathcal{S}_t|}\sum_{i\in\mathcal{S}_t}\sum_{k=0}^{K-1}\nabla\mathcal{F}_i(\theta_{i,k}^t) - \delta^t\|^2}_{(A)}$$

In (A), we take expectation with respect to $\mathcal{S}_t$ and total clients $N$. For that, we use Lemma 4 of (Reisizadeh et al., 2020). Specifically, using Eq. (59) in (Reisizadeh et al., 2020), we get:

$$(A) \leq \frac{\eta^2}{|\mathcal{S}_t|^2}\Big(\frac{|\mathcal{S}_t|}{N} - \frac{|\mathcal{S}_t|(|\mathcal{S}_t|-1)}{N(N-1)}\Big)\sum_{i\in[N]}\mathbb{E}\|\sum_{k=0}^{K-1}\nabla\mathcal{F}_i(\theta_{i,k}^t) - \frac{\delta^t}{\eta}\|^2$$

$$\leq \frac{\eta^2}{|\mathcal{S}_t|^2}\Big(\frac{|\mathcal{S}_t|}{N} - \frac{|\mathcal{S}_t|(|\mathcal{S}_t|-1)}{N(N-1)}\Big)\Big(2\sum_{i\in[N]}\mathbb{E}\|\sum_{k=0}^{K-1}\nabla\mathcal{F}_i(\theta)\|^2 + \frac{2}{N}\sum_{i\in[N]}\mathbb{E}\|\sum_{k=0}^{K-1}\nabla\mathcal{F}_i(\theta_{i,k}^t)\|^2\Big)$$

$$\leq \frac{4N}{|\mathcal{S}_t|^2}\Big(\frac{|\mathcal{S}_t|}{N} - \frac{|\mathcal{S}_t|(|\mathcal{S}_t|-1)}{N(N-1)}\Big)\frac{1}{N}\sum_{i\in[N]}\mathbb{E}\|\sum_{k=0}^{K-1} -\eta\nabla\mathcal{F}_i(\theta_{i,k}^t)\|^2$$

$$\leq \frac{4N}{|\mathcal{S}_t|^2}\Big(\frac{|\mathcal{S}_t|}{N} - \frac{|\mathcal{S}_t|(|\mathcal{S}_t|-1)}{N(N-1)}\Big)C^t$$

$$= \frac{4N}{(N-1)|\mathcal{S}_t|}(1 - \frac{|\mathcal{S}_t|}{N})C^t$$

This gives us the desired result. $\qquad\square$

**Lemma 8.** $I^t$ *in the proof of Lemma 3 has the following bound*

$$I^t \le 20\eta^2 K^3 \left(\delta_g^2 + B^2 \mathbb{E}\|\nabla\mathcal{F}\left(\theta_{i,0}^t\right)\|^2\right) + 10\eta^2 K^2 \sigma^2$$

*Proof.* Initially, we commence by deriving an upper bound for the variable $I_{i,k}^t$

$$
\begin{aligned}
I_{i,k}^t &= \mathbb{E}\|\theta_{i,k}^t - \theta_{i,0}^t\|^2 \\
&= \mathbb{E}\|\theta_{i,k-1}^t - \theta_{i,0}^t - \eta\nabla f_i\left(\theta_{i,k-1}^t\right)\|^2 \\
&= \mathbb{E}\|\theta_{i,k-1}^t - \theta_{i,0}^t - \eta\nabla\mathcal{F}_i\left(\theta_{i,k-1}^t\right) + \eta\nabla\mathcal{F}_i\left(\theta_{i,k-1}^t\right) - \eta\nabla f_i\left(\theta_{i,k-1}^t\right)\|^2 \\
&\le \mathbb{E}\|\theta_{i,k-1}^t - \theta_{i,0}^t - \eta\nabla\mathcal{F}_i\left(\theta_{i,k-1}^t\right)\|^2 + \eta^2\sigma^2 \\
&\le \left(1 + \frac{1}{K-1}\right)\mathbb{E}\|\theta_{i,k-1}^t - \theta_{i,0}^t\|^2 + K\eta^2\mathbb{E}\|\nabla\mathcal{F}_i\left(\theta_{i,k-1}^t\right)\|^2 + \eta^2\sigma^2
\end{aligned}
$$

The inequality in the fourth line is established due to the stochastic bounded variance assumption, while the inequality in the fifth line holds as a consequence of the Lemma 11. Now, let us proceed to determine the upper bound for $\mathbb{E}\|\nabla\mathcal{F}_i\left(\theta_{i,k-1}^t\right)\|^2$.

$$
\begin{aligned}
\mathbb{E}\|\nabla\mathcal{F}_i\left(\theta_{i,k-1}^t\right)\|^2 &= \mathbb{E}\|\nabla\mathcal{F}_i\left(\theta_{i,k-1}^t\right) - \nabla\mathcal{F}_i\left(\theta_{i,0}^t\right) + \nabla\mathcal{F}_i\left(\theta_{i,0}^t\right)\|^2 \\
&\le 2\mathbb{E}\|\nabla\mathcal{F}_i\left(\theta_{i,k-1}^t\right) - \nabla\mathcal{F}_i\left(\theta_{i,0}^t\right)\|^2 + 2\mathbb{E}\|\nabla\mathcal{F}_i\left(\theta_{i,0}^t\right)\|^2 \\
&\le 2L^2\mathbb{E}\|\theta_{i,k-1}^t - \theta_{i,0}^t\|^2 + 2\mathbb{E}\|\nabla\mathcal{F}_i\left(\theta_{i,0}^t\right)\|^2
\end{aligned}
$$

In the last inequality, we rely on the L-smoothness property of the local loss function for its validity. Substituting the derived upper bound for $\mathbb{E}\|\nabla\mathcal{F}_i\left(\theta_{i,k-1}^t\right)\|^2$ into our original expression,

$$
\begin{aligned}
I_{i,k}^t &\le \left(1 + \frac{1}{K-1} + 2K\eta^2 L^2\right)\mathbb{E}\|\theta_{i,k-1}^t - \theta_{i,0}^t\|^2 + 2K\eta^2\mathbb{E}\|\nabla\mathcal{F}_i\left(\theta_{i,0}^t\right)\|^2 + \eta^2\sigma^2 \\
&\le \left(1 + \frac{1}{K-1} + 2K\eta^2 L^2\right)I_{i,k-1}^t + 2K\eta^2\mathbb{E}\|\nabla\mathcal{F}_i\left(\theta_{i,0}^t\right)\|^2 + \eta^2\sigma^2
\end{aligned}
$$

Expanding the aforementioned recursion,

$$I_{i,k}^t \le \sum_{r=0}^{k-1}\left(2K\eta^2\mathbb{E}[\|\nabla\mathcal{F}_i\left(\theta_{i,0}^t\right)\|^2] + \eta^2\sigma^2\right)\left(1 + \frac{3}{K-1}\right)^r \le 10K\left(2K\eta^2\mathbb{E}\|\nabla\mathcal{F}_i\left(\theta_{i,0}^t\right)\|^2 + \eta^2\sigma^2\right)$$

Utilizing the above results, we can sequentially derive the upper bounds for $I_i^t$ and $I^t$.

$$
\begin{aligned}
I_i^t &= \sum_{k=0}^{K-1} I_{i,k}^t \le 10K^2\left(2K\eta^2\mathbb{E}\|\nabla\mathcal{F}_i\left(\theta_{i,0}^t\right)\|^2 + \eta^2\sigma^2\right) \\
&= 20\eta^2 K^3\mathbb{E}\|\nabla\mathcal{F}_i\left(\theta_{i,0}^t\right)\|^2 + 10\eta^2 K^2\sigma^2
\end{aligned}
$$

$$
\begin{aligned}
I^t &= 20\eta^2 K^3 \frac{1}{N}\sum_{i\in[N]}\mathbb{E}\|\nabla\mathcal{F}_i\left(\theta_{i,0}^t\right)\|^2 + 10\eta^2 K^2\sigma^2 \\
&\le 20\eta^2 K^3\left(\delta_g^2 + B^2\mathbb{E}\|\nabla\mathcal{F}\left(\theta_{i,0}^t\right)\|^2\right) + 10\eta^2 K^2\sigma^2
\end{aligned}
$$

The final inequality holds due to the assumption related to the averaged norm of the local gradient. This gives us the desired result.

$\qquad\square$

**Lemma 9.** $C^t$ in the proof of Lemma 3 has the following bound
$$C^t \le 20\eta^4 K^3 L^2 \sigma^2 + \left(40\eta^4 K^4 L^2 + 2\eta^2 K^2\right)\left(\sigma_g^2 + B^2 \mathbb{E}\|\nabla \mathcal{F}\left(\theta_{i,0}^t\right)\|^2\right)$$

*Proof.* Initially, we commence by deriving an upper bound for the variable $C_i^t$

$$
\begin{aligned}
C_i^t &= \mathbb{E}\left\|\sum_{k=0}^{K-1} -\eta\nabla\mathcal{F}_i\left(\theta_{i,k}^t\right)\right\|^2 \\
&\le K\eta^2 \sum_{k=0}^{K-1} \mathbb{E}\left\|\nabla\mathcal{F}_i\left(\theta_{i,k}^t\right)\right\|^2 \\
&= K\eta^2 \sum_{k=0}^{K-1} \mathbb{E}\left\|\nabla\mathcal{F}_i\left(\theta_{i,k}^t\right) - \nabla\mathcal{F}_i\left(\theta_{i,0}^t\right) + \nabla\mathcal{F}_i\left(\theta_{i,0}^t\right)\right\|^2 \\
&\le 2K\eta^2 \sum_{k=0}^{K-1} \left(\underbrace{\mathbb{E}\|\nabla\mathcal{F}_i\left(\theta_{i,k}^t\right) - \nabla\mathcal{F}_i\left(\theta_{i,0}^t\right)}_{L \text{ smooth}}\|^2 + \mathbb{E}\|\nabla\mathcal{F}_i\left(\theta_{i,0}^t\right)\|^2\right) \\
&\le 2K\eta^2 \sum_{k=0}^{K-1} \left(L^2\mathbb{E}\|\theta_{i,k}^t - \theta_{i,0}^t\|^2 + \mathbb{E}\|\nabla\mathcal{F}_i\left(\theta_{i,0}^t\right)\|^2\right) \\
&= 2K\eta^2\left(L^2 I_i^t + K\mathbb{E}[\|\nabla\mathcal{F}_i\left(\theta_{i,0}^t\right)\|^2]\right)
\end{aligned}
$$

The initial and fourth line inequalities are established based on Jensen's inequality. Using the aforementioned results, we can determine the upper bound for $C^t$.

$$
\begin{aligned}
C^t &= \frac{1}{N}\sum_{i\in N} C_t^i \le \frac{1}{N}\sum_{i\in N}\left\{2K\eta^2\left(L^2 I_i^t + K\mathbb{E}[\|\nabla\mathcal{F}_i\left(\theta_{i,0}^t\right)\|^2]\right)\right\} \\
&= 2K\eta^2 L^2 I^t + 2\eta^2 K^2 \frac{1}{N}\sum_{i\in N}\mathbb{E}[\|\nabla\mathcal{F}_i\left(\theta_{i,0}^t\right)\|^2] \\
&\le 20\eta^4 K^3 L^2 \sigma^2 + \left(40\eta^4 K^4 L^2 + 2\eta^2 K^2\right)\left(\sigma_g^2 + B^2\mathbb{E}[\|\nabla\mathcal{F}\left(\theta_{i,0}^t\right)\|^2]\right)
\end{aligned}
$$

Where the last inequality is established based on the assumption related to the averaged norm of the local gradient and Lemma 8. □

**Lemma 10.** $\sum_{t=1}^{T} \mathbb{E}\left\|m^{t-1}\right\|^2$ in the proof of Lemma 3 has the following bound

$$\sum_{t=1}^{T} \mathbb{E}\left\|m^{t-1}\right\|^2 \le \frac{T}{(1-\lambda)^2}\left(C_1\sigma^2 + C_2\sigma_g^2\right) + \frac{1}{(1-\lambda)^2}C_3\mathbb{E}\sum_{t=1}^{T}\left\|\nabla\mathcal{F}\left(\theta_0^t\right)\right\|^2,$$

where $C_1 = \eta^2 K^2 + 20\eta^4 K^3 L^2 A$, $C_2 = A\left(40\eta^4 K^4 L^2 + 2\eta^2 K^2\right)$, and $C_3 = B^2 A\left(40\eta^4 K^4 L^2 + 2\eta^2 K^2\right)$.

*Proof.* □

*First, let's unrolling the recursion of the momentum $m^t$, i.e., $m^t = \sum_{r=0}^{t}\lambda^{t-r}\Delta^r$.*

$$\mathbb{E}\left\|m^t\right\|^2 = \mathbb{E}\|\sum_{r=0}^{t}\lambda^{t-r}\Delta^r\|^2$$

*Let $\Gamma_t = \sum_{r=0}^{t}\lambda^r = \frac{1-\lambda^t}{1-\lambda}$. For $0 \le \lambda < 1$, $\Gamma_t \le \frac{1}{1-\lambda}$. Then,*

$$
\begin{aligned}
\mathbb{E}\|\sum_{r=0}^{t}\lambda^{t-r}\Delta^r\|^2 &= \Gamma_t^2\mathbb{E}\|\frac{1}{\Gamma_t}\sum_{r=1}^{t}\lambda^{t-r}\Delta^r\|^2 \\
&\le \Gamma_t\sum_{r=1}^{t}\lambda^{t-r}\mathbb{E}\|\Delta^r\|^2 \le \frac{1}{1-\lambda}\sum_{r=1}^{t}\lambda^{t-r}\mathbb{E}\|\Delta^r\|^2
\end{aligned}
$$

*Using above results, we can get*

$$
\begin{aligned}
\sum_{t=1}^{T} \mathbb{E} \left\| m^{t-1} \right\|^2 &\le \sum_{t=1}^{T-1} \sum_{r=1}^{t} \frac{\lambda^{t-r}}{1-\lambda} \mathbb{E} \|\Delta^r\|^2 \\
&\le \sum_{t=1}^{T-1} \sum_{r=1}^{t} \frac{\lambda^{t-r}}{1-\lambda} \left( \underbrace{\left( \eta^2 K^2 + 20\eta^4 K^3 L^2 A \right)}_{C_1} \sigma^2 \right. \\
&\quad + \underbrace{A \left( 40\eta^4 K^4 L^2 + 2\eta^2 K^2 \right)}_{C_2} \sigma_g^2 \\
&\quad + \left. \underbrace{B^2 A \left( 40\eta^4 K^4 L^2 + 2\eta^2 K^2 \right)}_{C_3} \mathbb{E} \left\| \nabla \mathcal{F} \left( \theta_0^r \right) \right\|^2 \right) \\
&\le \frac{T}{(1-\lambda)^2} \left( C_1 \sigma^2 + C_2 \sigma_g^2 \right) + \frac{1}{1-\lambda} \left( C_3 \sum_{t=1}^{T-1} \sum_{r=1}^{t} \lambda^{t-r} \mathbb{E} \left\| \nabla \mathcal{F} \left( \theta_0^r \right) \right\|^2 \right) \\
&\le \frac{T}{(1-\lambda)^2} \left( C_1 \sigma^2 + C_2 \sigma_g^2 \right) + \frac{1}{(1-\lambda)^2} C_3 \mathbb{E} \sum_{t=1}^{T} \left\| \nabla \mathcal{F} \left( \theta_0^t \right) \right\|^2
\end{aligned}
$$

The inequality in the second line can be obtained by substituting the result of Lemma 9 into Lemma 6. The inequality in the six line can be obtained by summing the coefficients corresponding to each $\left\| \nabla \mathcal{F}(\theta_0^t) \right\|^2$, respectively.

**Lemma 11.** *(Relaxed triangle inequality). For any $a > 0$, $\left\| v_1 + v_2 \right\|^2 \le (1+a) \left\| v_1 \right\|^2 + \left(1 + \frac{1}{a}\right) \left\| v_2 \right\|^2$*

*Proof.* This lemma holds because when we organize the formulas on the right, we get $0 \le \left\| a v_1 - \frac{v_2}{a} \right\|^2$. $\qquad \square$

