# OpenReview forum: "Communication-Efficient Federated Learning with Accelerated Client Gradient"
_ICLR.cc/2024/Conference — ICLR 2024 Conference Withdrawn Submission_

### Official Review · Reviewer_Nxpp · 2023-10-29

**Soundness:** 2 fair
**Presentation:** 3 good
**Contribution:** 2 fair
**Rating:** 5
**Confidence:** 3

**Summary:**

The authors propose a novel federated learning framework named Federated Averaging with Acceleration of Global Momentum (FedAGM), aiming to improve the stability of the server-side aggregation step. This is achieved by sending the clients an accelerated model estimated with the global gradient to guide the local gradient updates. The algorithm aggregates and conveys the global update information to participants without additional communication cost and does not require storing past models in the clients. The local update is regularized to reduce bias and improve the stability of local updates. The paper includes comprehensive empirical studies on real data under various settings, demonstrating the remarkable performance of the proposed method in terms of accuracy and communication-efficiency, especially with low client participation rates.

**Strengths:**

1. The proposed FedAGM algorithm is designed to be communication-efficient, which is crucial in federated learning scenarios where bandwidth and communication costs are significant concerns. The algorithm integrates the momentum into the global model, saving extra communication and computational costs. This is beneficial in large-scale and low-participation federated learning scenarios.
2. The paper demonstrates the effectiveness of FedAGM in terms of robustness and communication-efficiency in the presence of client heterogeneity. This is a strength as federated learning environments often involve diverse devices with varying computational capabilities and data distributions.
3. FedAGM improves the stability of the server-side aggregation step by sending clients an accelerated model estimated with the global gradient to guide the local gradient updates. This anticipatory update on the client side leads each client to find a local minimum adjacent to the trajectory of the global gradients, helping to avoid inconsistent local updates and improving the overall efficiency of the learning process.

**Weaknesses:**

The idea/motivation is not quite a surprise, which is based on the idea of Das 2020.
1. While the paper provides extensive evaluations and comparisons with several baselines, it might have been beneficial to include a wider range of existing personalized federated learning algorithms for a more comprehensive analysis.
pFedMe: Personalized Federated Learning with Moreau Envelopes Dinh et al., 2020
PerFedAvg: Personalized Federated Learning with Theoretical Guarantees: A Model-Agnostic Meta-Learning Approach Fallah et al., 2020
 APFL: Adaptive Personalized Federated Learning Deng et al., 2020
Ditto: Fair and Robust Federated Learning Through Personalization Li et al., 2022
2. The FedAGM algorithm involves several components and hyper-parameters  (e.g. lambda, eta), and while the paper does provide explanations for these, there might be room for further clarification and elaboration to aid readers in fully understanding the intricacies of the algorithm.
3. The effectiveness of FedAGM relies on the accurate estimation of the global gradient to guide the local updates. Any inaccuracies in this estimation could potentially affect the performance of the algorithm, and this aspect could be explored and addressed in more ablation study .

**Questions:**

Please see the Weakness.
Additionally, I would be curious the effects of tunning lambda and eta to see if they may create too much noise/bias to the model.

---

### Official Review · Reviewer_RTkn · 2023-10-30

**Soundness:** 3 good
**Presentation:** 3 good
**Contribution:** 2 fair
**Rating:** 5
**Confidence:** 3

**Summary:**

The paper proposes a solution to heterogeneous data and partial participation across clients in FL frameworks. In addition to server update momentum and local regularizer, their proposed algorithm employs sending an accelerated version of the model, which is the sum of the current global model and momentum term. The authors show the convergence guarantee of the proposed method under common assumptions. They experimentally show the superiority of their method over baselines in comprehensive experiments.

**Strengths:**

1.	The authors provide a method to aggregate past global gradient information for guiding local client training without additional communication cost. They show that their method theoretically guarantees the convergence at a matching rate with the literature.

2.	The experiments are quite comprehensive. Many methods in the literature are used as baselines to compare. The authors also provide extensive ablation studies.  Furthermore, the authors detailly explain the experimental setting.

**Weaknesses:**

-	I couldn’t fully understand the motivation of accelerated global model $\theta^{t-1}+\lambda m^{t-1}$. If this model is a better global model to send the clients for the local training, why don’t we update the global model and send it instead of just sending a modified global model? For example, changing the global model update line in Algorithm 1 (from $\theta^t\gets\theta^{t-1}+m^t$ to $\theta^t\gets\theta^{t-1}+(\lambda+1)m^t$)?
-	The acceleration at the global model is novel, however, the momentum and regularization in local training is similar to already existing methods, as also discussed by the authors in Appendix B.

**Questions:**

-	Some works (for example [1]) are proposed to decrease the variance while not requiring full participation, additional communication cost, or extra client storage that may be worth mentioning.
-	Regarding 1. in the weaknesses section, it may be good to include an ablation study where the global model is also updated to the accelerated global model ($\theta^{t-1}+\lambda m^{t-1}$), which is equivalent to rescaling the effect of $m^t$ in the global model update. It may be beneficial to show the effect of accelerated gradient.

[1] Jhunjhunwala, Divyansh, et al. "Fedvarp: Tackling the variance due to partial client participation in federated learning." Uncertainty in Artificial Intelligence. PMLR, 2022.

---

### Official Review · Reviewer_ufqp · 2023-10-30

**Soundness:** 2 fair
**Presentation:** 2 fair
**Contribution:** 2 fair
**Rating:** 3
**Confidence:** 3

**Summary:**

This paper tries to reduce client drift due to data heterogeneity among non-i.i.d clients. It introduces FedACG, a variant of FedAvg that incorporates a global momentum similar to Nesterov Accelerated Gradient(NAG) SGD . In each round, the server sends its momentum-integrated model to the clients as the initialization for that round. After receiving updates from the clients, it aggregates them to update the global momentum and the server model.

**Strengths:**

1) Data heterogeneity is an important problem for the federated learning (FL) community, which has received significant attention in recent years, and momentum based methods has proven effective for this problem.

2) Strong empirical results.

**Weaknesses:**

1) Novelty of the paper is limited, and the citations are improper.The difference of this paper and FedAvgM is that it uses Nesterov Accelerated Gradient(NAG) instead of Momentum SGD. However, the paper lacks citations related to NAG or a discussion of its relation to NAG.

2) The convergence bound is not optimal. The authors claimed their bound is $O(\frac{1}{\sqrt{SKT}})$, but due to some additive constants in the definition of A and $\psi_1$ the bound is actually $O(\frac{1}{\sqrt{T}})$. Even for the case of $\lambda=0$ which should be equivalent to FedAVG, the bound is not optimal.

**Questions:**

See above.